# Alternative Assisted Extraction Methods of Phenolic Compounds Using NaDESs

**DOI:** 10.3390/antiox13010062

**Published:** 2023-12-29

**Authors:** Mario Coscarella, Monica Nardi, Kalina Alipieva, Sonia Bonacci, Milena Popova, Antonio Procopio, Rosa Scarpelli, Svilen Simeonov

**Affiliations:** 1Department of Health Sciences, Università “Magna Græcia” di Catanzaro, Viale Europa, Campus Universitario “S. Venuta”, Germaneto, 88100 Catanzaro, Italy; mario.coscarella@studenti.unicz.it (M.C.); s.bonacci@unicz.it (S.B.); procopio@unicz.it (A.P.); rosa.scarpelli@studenti.unicz.it (R.S.); 2Institute of Organic Chemistry with Centre of Phytochemistry, Bulgarian Academy of Sciences, Acad. G. Bontchev Str. Bl. 9, 1113 Sofia, Bulgaria; kalina.alipieva@orgchm.bas.bg (K.A.); milena.popova@orgchm.bas.bg (M.P.); svilen.simeonov@orgchm.bas.bg (S.S.)

**Keywords:** bioactive compounds, environmental extraction, green chemistry, deep eutectic solvent

## Abstract

A renewed understanding of eco-friendly principles is moving the industrial sector toward a shift in the utilization of less harmful solvents as a main strategy to improve manufacturing. Green analytical chemistry (GAC) has definitely paved the way for this transition by presenting green solvents to a larger audience. Among the most promising, surely DESs (deep eutectic solvents), NaDESs (natural deep eutectic solvents), HDESs (hydrophobic deep eutectic solvents), and HNaDESs (hydrophobic natural deep eutectic solvents), with their unique features, manifest a wide-range of applications, including their use as a means for the extraction of small bioactive compounds. In examining recent advancements, in this review, we want to focus our attention on some of the most interesting and novel ‘solvent-free‘ extraction techniques, such as microwave-assisted extraction (MAE) and ultrasound-assisted extraction (UAE) in relation to the possibility of better exploiting DESs and NaDESs as plausible extracting solvents of the phenolic compounds (PCs) present in different matrices from olive oil components, such as virgin olive pomace, olive leaves and twigs, virgin and extra virgin olive oil (VOO and EVOO, respectively), and olive cake and olive mill wastewaters (OMWW). Therefore, the status of DESs and NaDESs is shown in terms of their nature, efficacy and selectivity in the extraction of bioactive phytochemicals such as secoiridoids, lignans, phenolic acids and alcohols. Related studies on experimental design and processes’ optimization of the most promising DESs/NaDESs are also reviewed. In this framework, an extensive list of relevant works found in the literature is described to consider DESs/NaDESs as a suitable alternative to petrochemicals in cosmetics, pharmaceutical, or food applications.

## 1. Introduction

As established by the development plan of the UN, there is a need for a more sustainable chemistry that looks after the environmental impact and sustainability of the methods employed. The new EU environmental policies and legislations for the period 2010–2050 [1] are calling, in fact, on a drastic reduction in solvents from non-renewable resources (e.g., fossil fuel): harmful volatile organic compounds’ use, although highly effective in many applications, must decrease [2,3,4,5,6,7,8,9,10,11,12]. High toxicity, high flammability, and non-biodegradability, being their main shortcomings, have exerted an intolerable pressure to the environment via unsustainable emissions. The green branch of chemistry, through green analytical chemistry (GAC), has investigated this challenge so far. Guided by 12 principles postulated in 2013 [13], GAC researchers have continuously strove to enhance the environmental performance from classical methods. As a result, greener options have emerged over petrochemicals. In particular, the new generation of ionic liquid (ILs), e.g., deep eutectic solvents (DESs) [14,15,16,17,18,19] and natural eutectic solvents (NaDESs) [20], are overtaking the way in many fields, as also witnessed in the scholar sector by an increasingly higher number of related publications each year. Indeed, DESs and their natural equivalents, NaDESs, have found a wide series of applications, including drug discovery [21,22,23,24] and drug delivery systems [25], employment as therapeutic deep eutectic solvents (THEDESs) [26,27], production of new nano-materials [28,29], desulfurization of fossil fuels [30], chromatography [31], organic synthesis [32], removal of environmental contaminants and separation of azeotropes [33], isolation and fractionation of compounds [34], and many more [35]. Likewise, in the food, cosmetic and pharmaceutical industry, following the view of the FAO for a circular economy based on zero waste, DESs and NaDESs are finding more and more space as a plausible means to extract bioactive compounds from several natural sources such as plants, vegetables, fruits and animals [36], as well as from different by-products and waste materials of the food and agri-food chain [37], with, to date, the majority of the studies on the discipline focused on the extraction of bioactive small molecules [38] and only a minority focused on the DESs’ extraction of bioactive biological macromolecules (e.g., proteins, carbohydrates, and lipids) [39].

Solvents designates as ‘green solvents’, such as subcritical water, supercritical fluids, ionic liquids, DESs and HDESs are all characterized by excellent properties, posing little to no toxicity to human health and to the environment, being more sustainable and bio-renewable in respect to the already existing hazardous [40]. The two latter however, which are often defined together with the term DESs [41] or sometimes also as deep eutectic liquids (DELs) [42] or as low-transition temperature mixtures (LTTMs), are the most promising and excel even over biomass-derived solvents [43]. Furthermore, their natural equivalents, i.e., NaDESs and HNaDESs, among many pros, for instance, also tend to more easily meet the GRAS (generally recognized as safe) requirements for solvents by the FDA [44], being that the chemical nature of NaDESs are fully compliant with the REACH Regulation. Moreover, it is accepted that DESs and NaDESs are less toxic than most organic solvents, and NaDESs are less toxic than DESs [45]. DESs have received increasing interest in a wide variety of chemical transformations [46,47]. In recent years, considerable attention has been paid to the application of DESs in the formation of amide bonds and in reactions for the protection of functional groups [15,16,17,18] (Figure 1). Furthermore, the possibility of creating new DESs for specific tasks has aroused great interest both in the field of organic synthesis and metals processing [48] and in biomass [49,50,51,52,53].

The peculiar and tailorable features of DESs, their versatility and ease of preparation, have made them gain a lot of attention especially in the extraction field of valuable compounds: their unique super-molecular structure manifests high affinity towards various classes of molecules, resulting in high solubilization power, extraction efficiency, and stabilizing ability [20,54,55,56,57].

One of the very first studies in which DESs were proven to extract various bioactive compounds was published in 2013 [20,58] and a first review taking green solvents into account for the extraction of PCs goes back to 2016 [59]. It was already clear, however, that there is no universal method valid for the extraction of all subclasses of plant polyphenols, considering the obvious heterogenicity of the category [60]. For the purposes of this review, attention will be paid to the extractability of DESs in regard to PCs from olive oil components. On this subject, one of the first papers was presented in 2016 by Garcìa et al. [61]. For the sake of truth, this review will also not consider the key findings/trends observed in DESs’ extraction of biomacromolecules, which are different from those observed in DESs’ extraction of small molecules [62].

This review will initially address the definition of DES and will then discuss, in a brief overview, the main advantages of them over ILs and their green solvent characteristics that govern their extraction performance. The main highlights in terms of analytical steps for the determination of bioactive compounds will also be resumed. A list of publications and their main key results in respect to PCs extracted using DESs/NaDESs, especially from matrices of olive oil components will then be reported. Thereafter, conventional extraction techniques will be compared to some novel ‘solvent-free‘ extraction methodologies, such as MAE and UAE. Finally, the dissertation will be concluded by drawing considerations and future perspectives aimed at aiding research in designing the best experimental conditions for the efficient extraction of PCs from olive oil components through the employment of DESs and NaDESs.

## 2. Overview of DESs

### 2.1. Definition of DESs

DESs have been a real breakthrough in the ground of green chemistry because of Abbott’s group publications [47,52,62,63]. The name ‘DESs’ takes its origin from ‘eutectic‘, namely as a mixture of compounds that, at a certain well-defined composition, displays a unique and minimum melting point in the phase diagram (Figure 1). The variation in the freezing point at the eutectic composition between a binary mixture of A and B and an ideal mixture can be quantified as ΔTf. This difference is directly influenced by the strength of the interaction between A and B. When the interaction is stronger, ΔTf will also be greater in magnitude. According to the most valued definition of the acronym DESs of Smith et al. [64], DESs (type I–IV) are salts formed by a eutectic mixture of Lewis or Bronsted acids and bases which can contain a variety of anionic and/or cationic species—type V of DESs are not necessarily made by salts but rather by molecular substances. Practically, DESs are formed by mixing, under certain optimal conditions (temperature and stirring time), two or three solid organic or inorganic compounds that do behave as hydrogen bond donor (HBD) and hydrogen bond acceptor (HBA) so that they liquefy (at a specific molar ratio) and form a stable eutectic. Depending on the type of the DES complexing agent, there are four/five types of DESs according to different authors: (I) quaternary ammonium salt with an anhydrous metal chloride; (II) quaternary salt with a metal chloride hydrate; (III) quaternary ammonium, sulfonium, or phosphonium salt (HBA) with an HBD compound; (IV) metal halide with an HBD; (V) ‘non-ionic DES‘ that are those in which both components are molecular substances [54,65]. In Figure 2, some common HBDs and HBAs in DES formation are reported; Table 1 summarizes instead the five mentioned DESs types.

With that being said, the most common type of DESs is type III, in which choline chloride (ChCl) constitutes the quaternary ammonium salt, acting as HBA and either urea, polyalcohols, sugars, amides, organic acids, or PCs are the HBD species. DESs type I and II are used to synthesize hydrophilic DESs, whilst DESs type III and IV are used for hydrophobic DESs [37]. ‘Natural‘ DESs or NaDESs are instead considered as being DES derivatives [41]. The term was coined to distinguish such a liquid made by primary or secondary metabolites of cells; this means that NaDESs are solvents prepared using natural components from cell metabolism. Over 135 primary metabolites based on NaDESs have been found, characterized by high polarity and hydrophilicity [66,67]. This has led to the hypothesis that the existence of natural DESs in living organisms might play an important role as a liquid phase for solubilizing, storing, or transporting non-water-soluble metabolites in living cells and organisms. Recently, another class of eutectic solvents has emerged for the extraction of phenolic compounds; these solvents are known as hydrophobic deep eutectic solvents (HDESs) and hydrophobic natural deep eutectic solvents (HNaDESs) [68,69,70], which are based respectively on, as HBA species, either quaternary ammonium salts with long alkyl chains or hydrophobic natural compounds, both coupled to hydrophobic HBDs such as carboxylic acids or alcohols with long alkyl chains, giving the solvent hydrophobic character. HDESs can be useful for the extraction of some PCs with nonpolar characteristics such as tocopherols [71]; HNaDESs have been employed for the purification of OMWW from endogenous phenol [70,72].

### 2.2. Brief Resumé on the Advantages of DESs over ILs

Research on DESs blew up in the early 2000s to attempt to overcoming the at least questionable and weak green character of ILs. In fact, although over conventional solvents, ILs have been exhibited several pros (such as negligible vapor pressure, good thermal properties, wide liquid range, wide solubility and miscibility range, suitability for chemical reactions, and good recycling properties), studies have also highlighted many cons (high preparation costs for large-scale applications that in some cases are ten times higher than for conventional organic solvents; similar or higher toxicity than organic solvents and low biodegradability) [73]. On the other hand, DESs are more inclined towards large scale-up processing, having much easier preparation and greater availability from relatively inexpensive raw material that does not require the tedious and costly dual-step synthesis of ILs that is not even devoid of by-products, while also exhibiting superior biodegradability, lower toxicity, chemical inertness with water, and fine tailorable properties. For these reasons, DESs are more properly considered a class of entirely newly generated fluids [74] rather than a subtype of ILs, as for many, the differences between the two outweigh the similarities. In fact, although DESs and ILs share similar physical properties, such as low volatility, high viscosity, chemical and thermal stability and non-flammability, they differ in the nature of the constituents, in the methods for the formation, and in the type of dominant intermolecular forces involved [75]. Here, we leave more detailed information on ILs to the attention of keen readers that can find details in the literature [76].

### 2.3. Green Characteristics of DESs

DESs and NaDESs share very similar physicochemical properties (strong ability to dissolve protic molecules, low vapor pressure, and miscibility with water, among others). As said, the most common DESs are type III formed by ChCl with cheap and safe HBDs, with the most popular ones being urea, ethylene glycol, and glycerol, but other alcohols, amino acids, carboxylic acids and sugars have also been quite often used [38]. DESs are characterized by a well-defined composition, which displays a significantly lower melting point in the solid–liquid phase diagram in respect to those of the pure compounds. The strong decrease in freezing point can span up to 200 °C, as in the case of the choline chloride:urea DES prepared at a molar ratio of 1:2 that reaches 12 °C, whilst the corresponding melting points are 302 °C (choline chloride) and 133 °C (urea) [47,52,62,63]. This interesting phenomenon of pure solid compounds that become liquid (at room temperature or at a temperature below 70 °C) by mixing them in a certain molar ratio under mild heat, according to the available instrumental analysis so far [77], is attributed to a decreased lattice energy due to hydrogen-bonding and van der Waals interactions formation [25,49]. For instance, in the case of the mentioned choline chloride:urea DES, the chloride ion of the HBA and the OH group of the HBD strongly link through hydrogen bonds—this interaction could explain the weakening of the lattice energy of the system from which the measured marked depression in melting point arises [78]. Having a solvent that is liquid at room temperature is a plus applicable to an extraction solvent. Another highlight is that the HBA-HBD molar ratio can influence the melting point. For instance, the choline chloride:urea DES at a molar ratio of 1:1 exhibits a melting point > 50 °C [34]. Also, the choice of the HBD partner has an important effect on the resulting melting point. For instance, the use of citric acid, malonic acid, oxalic acid, glycerol, ethylene glycol, and xylitol as HBDs resulted in the formation of DESs with melting temperatures of 69 °C, 10 °C, 34 °C, −40 °C, −66 °C, and room temperature, respectively [39]. More examples reporting the melting point of different combinations of DESs can be found in the review performed by Ling and Hadinoto [39]. For what concerns the production of DESs, this process is relatively straightforward and inexpensive and does not pose any significant post-purification or disposal problems. DESs can be conventionally obtained either by heating-stirring, grinding, evaporation, or freeze-drying. Usually, the components are put in a closed bottle and heated at 60 °C under stirring until a clear liquid appears. In the case of carboxylic acids as HBDs with ChCl, however, it is preferable to choose the grinding option over the heating one, or otherwise by-products (e.g., esters) from the reaction of the two species might form. So, with carboxylic acids, the DES components are rather pounded with a mortar to form the liquid to obtain a purer DES. The evaporation method, then, implies the dissolution of the DES components in water followed by evaporation at 50 °C with the rotary evaporator. The obtained liquid is put in a desiccator with silica gel until it reaches a constant weight [79]. Finally, the freeze-drying approach is less frequently used [80]. Apart from conventional ways, some greener preparation approaches have recently been introduced. Gomez et al. [81] prepared a microwave-assisted method that only needs 20 s of synthesis time to make the DES, while Santana et al. have presented an ultrasound-assisted method [82].

### 2.4. The Two Main Characteristics of DESs at the Basis of Their Extraction Performance: Viscosity and Polarity

The extraction performance of DESs is ruled by two principal properties: viscosity and polarity [83]. These two features can be tailored by the addition of water and changes in the extraction conditions (e.g., temperature, solid-to-solvent ratio, choice of the HBD species) [40,84].

The vast majority of DESs shows a higher viscosity at room temperature than many conventional solvents, but they are similar to ILs (>100 cP). The high viscosity of DESs is primarily due to the extensive hydrogen bonding type of interactions between each component of the system but, also, to a lesser extent, to van der Waals and electrostatic interactions [85]. This high parameter can be very beneficial when processing single drop micro-extraction but it limits the extraction applications of DESs in most settings, since it hampers the mass transfer rate between the sample and the extraction phase. Considering the inverse correlation between viscosity and extraction efficiency, it is more appropriate to choose low-viscosity DESs [39,59]. As a rule of thumb for DESs, the higher the number of OH groups present in the system, the more enhanced the hydrogen bond network and the higher the deriving viscosity. This results in limited mass transfer, thus reducing the extraction yield.

The inverse proportion between viscosity and temperature, on the other hand, can be useful in promoting extraction efficiency. In fact, since viscosity decreases significantly when the temperature increases, by raising the temperature up to a certain limit, the internal resistance of molecules decreases, causing the molecules to flow more easily [71]. For instance, the viscosity of glucose:ChCl:water is decreased by 2/3 when the temperature increases from 20 °C to 40 °C. With that being said, a too elevated raise in temperature levels might also impact the chemical bonds and structure of the targeted compounds, leading to thermal degradation and/or oxidation of targeted phenolics, consequently reducing the extraction efficiency [86].

Another option for lowering the viscosity of DESs is represented by playing with the HBD component. For instance, DESs have higher viscosity when sugars and carboxylic acids are employed as an HBD partner, whilst ethylene glycol, glycerol, and phenol as HBD species result in less viscous DESs. As already explained, the higher the number of OHs, the stronger the hydrogen bonding interaction, the higher the viscosity, and the worse the extraction efficiency.

Research has also discovered that adding water to DESs is able to diminish their viscosity. Upon water addition, the weakening of the hydrogen bonding and the increase in osmotic pressure enhance the mass transfer, thus lowering viscosity. This effect stays optimal when the water content is generally in between 20% and 30% (*v*/*v*). Above that value, i.e., if the water content goes above 50% (*v*/*v*), the hydrogen bonding between the species is so weakened that it even disappears and the DES loses its stability and exists only as a liquid with individual and separated HBA and HBD [87,88].

Aside from viscosity, the other strategic factor for improving the extraction efficiency of DESs is polarity [89]. Indeed, having a solvent with a polarity close to the one of the desired compounds to extract favors solubilization, ultimately empowering the extraction efficiency [90]. The polarity of DESs increases with an increasing proportion of water. This effect was already visible in one of the first studies of Choi et al. [20]. Also, the polarity of DESs varies when different HBD components are used. Organic acid-based DESs are reported to be the most polar (44.81 kcal/mol) and both sugar- and polyalcohol-based DESs are less polar, with a polarity value closer to that of methanol (51.89 kcal/mol) [91]. It is also possible to adjust the polarity of a DES by tailoring the molar ratio of the DES components.

A more comprehensive and detailed list of studies in which the extraction efficiency of DESs is correlated to important analytical parameters can be found in the study of Ali Redha [84] and in the study of Ling et al. [40]. Together with the findings they present, it is possible to draw some conclusions. The viscosity of DESs should be low, otherwise the solubility of the targeted compounds might be not so favorable, resulting in poor extraction efficiency. Change in water content and mild temperature increase can hamper the high viscosity of DESs and facilitate the extraction process. The HBD component can influence the physicochemical properties of DESs. For these reasons, an evaluation of the properties is therefore necessary to ensure optimized extraction performance. Nonetheless, extraction process variables, such as extraction temperature, time, and liquid-to-solid ratio, also play critical roles in the extraction efficiency of target compounds. Lastly, advanced technologies like microwave and ultrasound can boost the extraction performance.

## 3. General Analytical Procedures for the Determination of Bioactive Compounds from Plant Matrices

It is now opportune to summarize the basic main steps for the determination of bioactive compounds from various natural sources—as the aim of this review is to then consider the possibility of employing DESs as a tool for the extraction of PCs from olive oil components through innovative extraction techniques. They can be listed as it follows: sampling, sample pretreatment, extraction of the targeted compounds, purification of crude extracts, qualitative identification, quantitative determination, and the statistical and chemometrical analysis of the results—the latter two are listed here but will not be illustrated afterwards.

Recently, a detailed recommended protocol for the selection and characterization of plant material was recently published in a review by Kellogg et al. [92]. It offers guidance for the effective selection of botanical natural products for research purposes. In fact, although it is stated that this process of selection must be tailored on a case-to-case basis, it is also desirable to design rigorous and reproducible studies since in the beginning of any scientific investigation; adhering to the guidelines is good laboratory practice (GLP) and we want to highlight that practical research requires the selection and optimization of each individual step of the entire analytical procedure.

Natural products differ from their pharmaceutical counterparts in that they are typically complex mixtures, for which the identities and quantities of the components present are not fully known. The composition of natural products can vary depending on the method of preparation or source material used. For sampling concerns, notwithstanding that it basically relies on the aims and scopes of the investigation, it is necessary to determine which part or portion of the selected plants should be sampled and when sampling should be performed [93]. Afterwards, the representative number and type of plants for sampling and the sampling procedure (random or selected) must be determined [94]. With the expansion of knowledge, for instance, a list of samples to exclude prior to chromatographic analyses is available (plants which have been mechanically injured, plants that are diseased, plants exposed to stresses, and so on [95,96]). Also, it is clear that the specific cultivar selected for the study can be a carrier of unique chemical structures and compositions of precious compounds through its genotype and geographical area [97].

In addition, sample pretreatment is one of the key steps of the qualitative and quantitative analysis of bioactive compounds. This procedure might be specified in the exact analytical protocol chosen for the plant matrix used and for the specific extraction method employed each time or it might be otherwise adjusted and varied in compliance with the requirements of the experiment. In both cases, through several options, it can allow the enhanced release of polyphenols, which are located within the cytoplasm and vacuoles of the cell, into the extraction media, thus boosting extraction efficiency. As anticipated, pretreatment depends on the matrix type of the particular sample—nowadays, bioactive compounds are extracted from diverse samples of plant matrices, including medicinal plants [98], fruits and vegetables [99], edible flowers [100], nus-products from various industries (especially the food one) [101,102,103,104,105], and so on. Sample pretreatment can be accomplished through several stratagems, with either biological, chemical, or physical agents, depending on the case. Usually, before the extraction, plant samples are simply treated by milling, grinding, and homogenization, which may be preceded by air-drying or freeze-drying [106]. These practices are useful to increase the surface area for proper mixing with the solvent and partially release the components trapped with the plant cell, thus favoring the effectiveness of the extraction process itself. However, there are some precautions to keep in mind in order to reach an optimal extraction. In the case of grinding, for instance, a too small particle size would cause difficulty in separating the extract from the residues and additional clean-up steps may be required. In these cases, either separation techniques or centrifugation can be employed. The clean-up procedure however, might be useful in any case to deprive the sample from impurities, cellular debris, and unwanted cellular fractions as lipids and chlorophyll (the latter is one pro-oxidant activator that can alter measurements of antioxidant activity for instance) prior to the extraction itself [107]—elimination of lipids and chlorophyll can be achieved by soaking the sample overnight with an organic solvent as petroleum ether [108]. Apart from solvent pretreatment that acts as an ‘extractant’ [109], which is also sometimes achieved by soaking the sample with a mixture of ethanol and water, the enzymatic pretreatment is another choice. Indeed, the pretreatment of olive leaves by cellulase in the extraction of hydroxytyrosol by NaDESs was found to be effective [110]. Considering that enzymes can break down and dissolve the cell wall (which is primarily composed of cellulase and other lignocellulosic content) of the cells of the natural source, their utilization as a pretreatment method can be considered as very effective [111,112].

With that being said, it is noticeable, however, that innovative extraction techniques, coupled to DESs as solvents, can already provide a high extraction efficiency without the need of excessive pretreatment. In fact, for example, in the case of MAE, most DESs, being polar, have a high dielectric constant and can absorb microwave energy which is converted into heat and transferred in turn to the cell wall that breaks up and releases analytes into the solvent matrix. During the MAE process, the dehydration of cellulose and the subsequent swelling of plant cells cause further evaporation of moisture which contribute, in turn, to the increased mass transfer of solutes. This weakens the mechanical strength until cells disrupt with the resulting phytochemical leaching [113,114,115].

Regarding the extraction process of the targeted compounds, the extraction technique (with a specific analytical extraction protocol which also depends on the type of the matrix) and the solvents used are the nitty-gritty of the matter. Other variables, however, such as the chemical and physical properties of the target compounds, extraction time, temperature, pH, and type of matrix have a relevant influence as well [76]. The old-fashioned way of achieving extraction will be presented elsewhere: it comprises techniques such as maceration, digestion, decoction, infusion, percolation, and Soxhlet extraction [92]. In relation to the extraction of phenolic compounds, many studies have appealed to typical solid−liquid extraction (SLE) and heat refluxing extraction (HRE). Some of the shortcomings of the conventional methods can be overcome by more effective and innovative techniques, such as microwave-assisted extraction (MAE), ultrasound-assisted extraction (UAE), enzyme-assisted extraction (EAE), high intensity pulsed electric field (HIPEF), pressurized liquid extraction (PLE), homogenate-assisted extraction (HAE), and high hydrostatic pressure-assisted extraction (HHPAE), various liquid-phase microextractions (LFMEs), etc. [116]. The most widely used of these innovative techniques for the extraction of phenolic compounds with DESs/NaDESs are MAE, UAE, HAE, and HHPAE [84]. In this review, as said, emphasis will be placed especially on MAE and UAE.

When it comes to the purification steps needed to recover purified compounds from crude extracts after the extraction with DESs/NaDESs has been carried out, it is noted to keep in mind that DESs and NaDESs are solvents characterized by very low vapor pressure. This aspect is considered an asset during extraction because the temperature can be increased without large evaporation of the solvent but it can even be a drawback for non-volatile solute evaporation. There are several strategies to recover PCs from DESs: solid–liquid extraction (SLE) using microporous resins or molecular sieves, liquid–liquid extraction (LLE) using another solvent, precipitation by the addition of anti-solvents (water preferentially), back extraction, supercritical CO_2_-based extraction and preparative high-performance liquid chromatography (prep-HPLC). All these methods are more extensively described in the work of Durand et al. [79] and in the work of Della Posta et al. [45]. With that said, to some, it is arguable to provide the means for the removal of the solvent. This line of thought might find a match in the case of the DES containing extracts could eventually be directly used without a solvent removal process, especially when the usage of the DES has enhanced the biological activity. For instance, the study reported by Ling et al. [117] showed that the formulated DES enhanced the solubility and antioxidant capacity of antioxidant extracts by up to 15% and 14.64%, respectively. Further studies and the end-use of each investigation might justify or discard the need for this practice.

Before proceeding with a brief description of the quantitative and qualitative analysis, it is also convenient to bear in mind that, even prior to the purification of crude extracts, there might be some intermediate steps for mandatory clean-up procedures. This can be true for some very sensible techniques to remove interfering material that might even irreversibly damage their delicate apparatuses. For example, before running a HPLC of the extracts, in the work of Chan et al. [107], which provides a series of guidelines and references for MAE of bioactive compounds in plants, the collected extracts are centrifuged at a speed ranging from 400 up to 8000 rpm from 5 to 15 min and filtered through a 0.45 μm membrane. For some thermo-stable and non-volatile active components, the extract is usually concentrated by a rotary evaporator at 45–65 °C under reduced pressure and filtered prior to quantification analysis.

With respect to the identification and quantification of extracted bioactive compounds, there is a wide window of methods with different combinations. For simplicity and low costs, UV-Vis spectrophotometric methods are the most used [118]. These methods lack selectivity towards single analytes and can only be applied to samples containing compounds able to absorb radiation at the wavelength of the light source—the technique can instead give satisfactory results if, for instance, the standard compounds are available for structural confirmation of target analytes. Nonetheless, UV-Vis methodologies allow the determination of the total content of some subclasses, i.e., total phenolic content (TPC), total flavonoid content (TFC), total anthocyanin content (TAC), and/or total antioxidant capacity expressed as iron (III) reducing capacity (FRAP), 2,2-diphenyl-1-picrylhydrazyl radical assay (DPPH), and 2,20-azino-bis-(3-ethylbenzothiazolin-6-sulfonic acid) radical assay (ABTS) [119]. The gold standard method for the unambiguous identification of single PCs in a mixture and their chiral isomers is the nuclear magnetic resonance spectroscopy (NMR), as also demonstrated for PCs extracted from olive oil without the need for sample pretreatment [120]. However, due to the investments and running costs of the device, for routine analyses, its use is not intended and justified, as, in fact, more powerful and selective techniques are more commonly used, such as thin-layer chromatography (TLC), liquid chromatography, and/or high-performance liquid chromatography (LC/HPLC) and gas chromatography (GC). Traditional TLC and its recent variant high-performance thin-layer chromatography (HPTLC) are very popular [121,122]: also, the combination of the powerful separation of TLC/HPTLC with a mass spectrometer (MS) as a detector leads to the unambiguous identification of individual compounds from different bioactive classes [123,124,125]. For the sake of money and for simplicity, the connection of LC/HPLC devices to different detectors, as ultraviolet-visible (UV-ViS), MS, evaporative light scattering detector (ELSD), and electrochemical detector (ECD) makes one of these combinations the most preferred choice [125]. It is not then in the aims of this review to further deepen the argument on devices and settings available for identification and quantification purposes, for it would require much more attention. Additionally, we remember that HPLC is not a suitable technique for the quantification of volatile bioactive compounds (e.g., essential oils) and the use of GC is a proper choice in this case.

## 4. DESs in the Extraction of Olive Oil PCs

### 4.1. Olive Oil Is One of the Pillars of the Mediterranean Diet

The Mediterranean Diet, in 2020, celebrated its 10-year anniversary of its inscription in the UNESCO Representative List of the Intangible Cultural Heritage of Humanity [126]. One of its most acknowledged and virtuous components thought to be responsible for its health benefits is olive oil. Indeed, olive oil is a functional food that has gained a health claim from the European Food Safety Authority (EFSA) [127]. The evidence from human studies has shown significantly reduced levels of oxidized low-density lipoprotein (LDL) in plasma after virgin olive oil consumption; the EFSA has stated the following: ‘olive oil polyphenols contribute to the protection of blood lipids from oxidative stress’ and ‘the claim may be used only for olive oil containing at least 5 mg of hydroxytyrosol and its derivatives (e.g., oleuropein complex and tyrosol) per 20 g of olive oil’. These polyphenols mentioned in the EFSA claim conferring such properties belong to a larger ensemble of soluble compounds that, together with the unsaponifiable part of the oil, constitute the so-called ‘minor- or micro-constituents’ of the olive oil. These microconstituent comprehend a plethora of at least 230 chemical compounds, with the majority being aliphatic and triterpene alcohols, sterols, hydrocarbons, and antioxidants such as carotenoids and polyphenols, which are responsible for the organoleptic properties, stability, and nutritional value of extra virgin olive oil (EVOO) [128,129]. Olive oil is a source of at least 30 PCs [130,131,132,133,134,135,136,137,138,139,140,141]. PCs, in olive oil (Figure 3), do occur as phenolic acids or alcohols (or phenylethanoids), secoiridoids as aglycons and oleuropein derivatives, lignans, and flavonoids. With that said, the major phenolic compounds in olive oil are the secoiridoids oleuropein (OLE), demethyl-OLE, and ligstroside, with their hydrolysis derivatives, such as their aglycons, oleuropein aglycone (3,4-DHPEA-EA: 3,4-dihydroxyphenylethanol-elenolic acid monoaldehyde), oleacein (3,4-DHPEA-EDA: 3,4-dihydroxyphenylethanol-elenolic acid dialdehyde), and oleocanthal (p-HPEA-EDA: *p*-hydroxyphenylethanol-elenolic acid dialdehyde) and their phenolic acids, i.e., hydroxytyrosol (3,4-DHPEA or HT) and tyrosol (p-HPEA or TY)—Figure 3 reports the main polyphenols present in olive oil; for a more detailed chemical classification of PCs in olive oil, see the work of Finicelli et al. [142]. These major PCs are bioavailable in humans and, through pleiotropic mechanisms of action, exert in vivo antioxidant roles [143], with potential anti-inflammatory and antiproliferative effects—many of the measured in vitro effects still require validation by high-quality human trials [144].

The amount of PCs in olive oil ranges from 20 to 1400 mg/kg, depending on agronomic factors (like the variety of the olive), on the ripeness of olives, as well as on the extraction technology, along with the storage and packaging processes [152]; in EVOO, for instance, PCs concentration can be low (20–200 mg/kg), medium (450–700 mg/kg), and high (750–1400 mg/kg) [153]; in virgin olive oil, it is usually around 230 mg/kg, with a common range of 130–350 mg/kg [154]. Interestingly, around 80% or more of the olive oil PCs are lost in the refining process. Most definitely, PCs, following industrial processing, considering their hydrophilic character, end up in relevant concentrations in waste material (e.g., olive mill wastewaters (OMWW), leaves and crude extracts, pomace, olive cake and the pits discharged from EVOO) [155,156]. Indeed, it is known not only that PCs are highly present in olive fruits but also, surprisingly, that olive oil, as a final product, contains only 2% of the total phenolic content of whole olives, while the remaining 98% ends up in OMWW [157,158]; for a comprehensive description of the compounds present in different elements of *Olea aeuropea* L., see the review of Ghanbari et al. [159]. As in the case of OMWW, owing to being a pollutant with a high chemical oxygen demand (COD) [160], some of these wastes pose a significant threat to the environment. For all that has been said so far, the recovery and revalorization of PCs from olive oil by-products such as OMWW, oil cake, oil pomace and leaves for nutritional and pharmaceutical purposes is of great attractiveness. In the next section, a list of studies that have used DESs coupled with innovative extraction techniques for extracting PCs from several olive oil components, including some waste materials, will be reported, discussed, and summarized in Table 2.

### 4.2. Comprehensive List of Literature Studies with DESs Used in Olive oil PCs Extraction

In this section, relevant publications from 2016 to present and their main key results/findings in respect to PCs extracted using DESs/NaDESs from different matrices of olive oil components with either conventional or innovative extraction techniques are reported in chronological order. To narrow down the field to the molecules mentioned in the previous paragraph, only results concerning PCs will be reported and discussed; results on related classes of compounds, as flavonoids, even if structurally close to PCs, will not be mentioned. Table 2 reports the experimental conditions and the main relative data of these experiments.

For the first time in 2016, in a study by Garcìa et al. [61], DESs have been applied for the extraction of PCs from VOO with a liquid-phase microextraction technique (LPME). Different DESs consisting of ChCl in association with sugars, alcohols, organic acids, and urea, as well as a DES of three sugars, were used. The PCs recovery from DESs was carried out by nonpolar resin XAD-16 (styrene divinylbenzene)—the official method for the extraction of phenolic compounds from VOO by the International Olive Council Testing Method uses 80% (*v*/*v*) methanol at room temperature and the removal of organic solvent for the concentration of polyphenols from the extracts is accomplished by low-temperature evaporation. In the study of Garcìa et al. [61], 14 gr of different DESs were at first mixed in a water bath with 14 gr of VOO samples under agitation. The two resulting phases were centrifuged at 1200× *g* for 10 min. The lower phase (polar extracts) was washed with hexane until, upon decantation, two phases appeared. Again, the polar extract was washed and bubbled with nitrogen. The obtained DES extracts were passed through the column. PCs were recovered by flushing the column with 100 mL of MeOH. The methanolic samples were evaporated under vacuum at 35 °C and redissolved with 2 mL of MeOH for analysis via HPLC-DAD-MS. A quantitative analysis of the ten most characteristic peaks of the chromatograms revealed that the two most abundant dialdehyde secoiridoid derivatives, oleacein, and oleocanthal, extracted with ChCl:xylitol (2:1) and ChCl:1,2-propanediol (1:1) showed an increase of 33% and 68% with respect to conventional extraction (80% (*v*/*v*) methanol/water), respectively. Concerning oleuropein aglycon (Hy-EA) and ligstroside aglycon (Ty-EA), the extraction with ChCl:1,2-propanediol (1:1) and ChCl:1,4-Butanediol (1:5), respectively, showed a 48.4% and 156.6% increase compared to the control (conventional methanol extraction carried out at room temperature). ChCl:1,2-propanediol (1:1) was found to be one of the most efficient solvents, able to extract less polar compounds, the flavonoids (luteolin and apigenin), and lignan (acetoxipinoresinol), in agreement with the results reported by Choi et al. [20] for the extraction of PCs from safflower Carthamus tinctorius. In relation to the sugar-based DES, ChCl was used to prepare two DESs, ChCl:sucrose (1:1) and ChCl:sucrose (4:1)—of which to the former was added 25% of water to reduce viscosity and improve extraction efficiency that was manifested for almost all of the PCs. Also, a xylitol-based DES, ChCl:xylitol (2:1), was prepared with the same amount of water addition. It showed a high capacity for extracting phenolic compounds from olive oil. Finally, a NaDES, i.e., a DES of non-toxic primary metabolites was made. The fructose, glucose, and sucrose DES (1:1:1) only showed high efficiency for the extraction of Hy-Ac and oleacein; nonetheless, it remained very interesting for its eco-friendly composition. In conclusion, the order of the extraction efficiency over PCs as the sum of the total PCs from olive oil was ChCl:xylitol (2:1) > ChCl:1,2-propanediol (1:1) > ChCl:glycerol (1:2) > control sample ≈ ChCl:sucrose (1:1) > 80% (*v*/*v*) methanol/water sample > ChCl: 1,4-Butanediol (1:5) > fructose:glucose:sucrose (1:1:1) > ChCl:Malonic acid (1:1) > ChCl:lactic acid (1:2) > H_2_O > ChCl:sucrose (4:1). The urea-based DESs showed the least results over HT and TY and showed comparable outcomes over oleacein and oleocanthal in respect to some of the other DESs. The authors have hypothesized that the solubilizing abilities of the DESs seen essentially depend on the relative capacities of forming hydrogen bonding interactions with the targeted compounds.

A successful evaluation of the properties of a fine-tailored DES in extracting PCs from olive leaves (OLL) was carried out by Athanasiadis et al. [161]. They used samples of some dry Greek olive leaves powder mixed with 25 mL of aqueous DES and stirred at 600 rpm for 120 min at 50 °C. Extraction with glycerol:glycine:water (7:1:3) DES afforded almost 18% higher total polyphenolic yield than aqueous ethanol (AE), which was over than 24% higher than aqueous methanol (AM) and approximately 29% higher than water. The DES extract also displayed stronger antioxidant effects. Thus, this DES was identified as a means to boost polyphenol recovery.

Athanasiadis et al. [162] have used the previously identified glycerol:glycine:water (7:1:3) DES as aqueous solution 80% (*w*/*v*) to boost the extraction of PCs from Olea europaea leaves in the presence of methyl β-cyclodextrin (CD). Indeed, cyclodextrins can form stable inclusion complexes with sparingly water-soluble molecules (such as polyphenols), increasing their solubility [163]. In this case, olive leaves powder was mixed with CD and the extraction was performed under the stirrer for 180 min. Samples were then centrifuged at 20,000× *g* to collect the clear supernatant after dilution of 1:20 with water. The clear solutions were then characterized via LC-DAD-MS. Results showed that CD significantly enhanced the total polyphenolic yields of the DES/CD (+17.8%) compared with the aqueous ethanol, proving for the first time the concept of β-cyclodextrin-aided polyphenol extraction in combination with a DES. The DES’ extracts also displayed stronger antioxidant capacity compared with AE [164].

TY and HT extraction from Croatian olive pomace samples has been performed through an ultrasound-microwave cooperative reactor by Panić et al. [165]. The chosen DES was ChCl:citric acid (1:2) with 30% water added (*v*/*v*), for it was characterized by low pH value and polarity similar to those of water and polar organic solvents used for PCs’ extraction. The extracts, after centrifugation at 5000 rpm for 15 min and recovery of the supernatant, were analyzed by means of HPLC. Identification and quantification through external standards have led to amounts of analytes expressed as mg of compound per kg of dry weight (DW) of olive pomace. The quantity of TY recovered with the mentioned DES (136.7 mg·kg^−1^ DW of pomace) was almost double of the figure obtained using ethanol (69.5 mg·kg^−1^ DW of pomace); for HT, only the value using the DES was reported as 195.6 mg·kg^−1^ DW of pomace.

In the study of Paradiso et al. [166], an easy and green method for labeling olive oils based on their PCs content was set up through a liquid–liquid extraction procedure based on a NaDES and direct spectrophotometric analysis of the extract. Lactic acid, glucose and water were mixed (3:1:3), resulting in a NaDES with a pH of 1.24 to whom 35% of water was added to reduce viscosity. The effect of intense vortex agitation followed by centrifugation for 10 min at 6000 rpm has been evaluated on the mixture of 1 g of oil added with 1 mL of hexane and 5 mL of NaDES. The lower layer containing NaDES plus phenolics was recovered, centrifuged again at 9000 rpm for 5 min, and recovered again. Finally, it was filtered at 0.45 μm using nylon filters. Subsequent HPLC separation, spectro-photometrical, and statistical analyses allowed the determination of the content of PCs with a mean error of 35.5 mg kg^−1^, correctly labeling 98.2% of the 163 olive oils samples according to the legal requirements for the health claim of the EFSA.

Buldo et al. [70] used a liquid–liquid extraction technique to detoxify OMWW from endogenous phenol, which is highly harmful to humans. They used three HNaDESs: octanoic acid:dodecanoic acid (3:1)—the former acting as HBD and the latter as HBA—and two DL-menthol-based NaDES—DL-menthol acting as HBA. These menthol-based NaDES in the ratio 1:2 were lauric acid:DL-menthol and caprylic acid:DL-menthol; the two acids were made of, respectively, C_8_ and C_12_. It is interesting to notice that C_8_:C_12_ HNaDESs are hydrophilicity-switchable by simply altering the pH by putting them in contact with an aqueous solution of a weak amine [167]. The extraction was carried out by mixing appropriate amounts of a pH-conditioned synthetic water-based matrix (reflecting the typical composition of measured OMWW) and of the adopted HNaDESs. After the water phase was sampled, it was filtered through a cellulose acetate membrane filter (0.20 μm) and analyzed via HPLC. The results showed that, as is expected for molecules of the toxic phenol whose solubility is essentially dominated by the hydrophobic core of the aromatic ring, the reduced dissociation causes reduced solubility in water and increased solubility in hydrophobic substances alike, such as the HNaDESs used. This reflects in the fact that all three adopted HNaDESs show maximum selectivity in extracting phenol rather than TY from the water phase containing both, as a function of the pH—strangely, the maximum selectivity was seen at a neutral pH, which is a fact that that needs further in-depth investigation. Nonetheless, being the most hydrophobic, the menthol-devoid HNaDES, i.e., octanoic acid:dodecanoic acid (3:1), was indeed the most favorable solvent in accommodating phenols, which is only poorly solubilized by OMWW at all pH values. The results showed that HNaDESs are a promising food-safe solvent to improve existing processes for the separation of the healthy fraction of polyphenols from OMWW.

Francioso et al. [168] developed a two-step green method with the aim to extract, isolate and purify HT, TY, oleacein and oleocanthal from EVOO. The DES-based extraction was carried out by adding ChCl:glycerol (1:1.5) to the oil in a ratio of 1:20 (*w*/*w*). The extraction was repeated three times and PCs were quantified via UPLC-DAD/MS analysis to evaluate the efficiency of the extraction and the relative recovery of the single compounds after each step; the first extract, containing HT, TY, oleacein and oleocanthal, was subjected to HPLC-preparative analysis to purify the individual bioactive molecules while the second and third extracts were placed together before prep-HPLC analysis. A reversed-phase chromatography column was employed for preparative separation, selecting an elution performed with a binary gradient system considering as mobile phase water and 80% ethanol. The method allowed the quantitative recovery of analytes found in the extracts.

Kaltsa et al. [169] have set up a method to produce polyphenol-enriched extracts using a L-lactic acid:glycine DES (5:1) tested as 70% (*w*/*v*) aqueous mixture by starting from dried Greek olive leaves. Extraction of dried leaves with the described DES was carried out under stirring at 50 °C with a heated oil bath for 150 min, followed by centrifugation at 10,000× *g* for 10 min. Extracts were also analyzed via LC-DAD-MS and HPLC. The measured total polyphenol yield was 93.73 mg of gallic acid equivalents (GAE) g^−1^ DM for the DES extract. The value almost doubled the one obtained by using water as solvent and was also significantly higher than the ones obtained either with methanol or ethanol 60%. Interestingly, the value of HT in the DES extract was six times higher than the one in the methanolic extract, demonstrating the OLE hydrolysis yielding to HT during the 150 min of the extraction. This finding indicated that the DES used might be a benign and effective means of OLE hydrolysis to produce HT, which might be a more potent antioxidant.

Kurtulbas et al. [42] have tested 23 combinations of DESs containing either salts (zinc chloride and potassium chloride), amino acid (L-proline), organic acids (citric acid, oxalic acid, malonic acid, and malic acid), polyalcohols (xylitol and glycerol), sugars (sucrose and maltose), and amines (urea, dimethyl urea, and N-methyl-urea) to extract OLE from Turkish olive leaves through digital homogenization. The method coupled with further electrochemistry analyses also allowed the detection of trace amounts of OLE. Olive leaves were extracted three times with the designed DESs through homogenization (10,000 rpm for 60 s). After the extracts were collected, the total mixture was centrifuged at 5000 rpm for 25 min. Before HPLC analysis, the extracts were filtered through a syringe filter (0.45 µm) and kept in the dark at −20 °C. L-proline:oxalic acid (1:4) turned out to be the best extracting solvent leading to the highest OLE yield seen (15.66 mg·g^−1^ DM corresponding to 224 mg/L) that surpassed the one obtained by the control extraction with water (168 mg/L). The total biophenol concentration measured through spectrophotometric analysis using the mentioned DES as a solvent for the extraction was 41 mg of GAE·g^−1^ DM. The result of the total biophenol concentration was in line with the previous measurements of the group that were in between 10.11 and 61.66 mg of GAE·g^−1^ DM [170].

Chakroun et al. [171] have recovered olive leaf PCs using a L-lactic acid/ammonium acetate DES composed of 54.6% (*w*/*w*) with 0.7% (*w*/*v*) β-cyclodextrin (β-CD). Aliquot of 0.57 g of olive leaf powder was extracted with 20 mL of the described DES and the extraction was undertaken under stirring for 150 min at 50 °C. The extract was centrifuged at 10,000× *g* for 10 min and the supernatant was spared for LC-DAD-MS analysis. The maximum polyphenols extraction yield, which was expressed as mg of caffeic acid equivalents (CAE) g^−1^ DM, was 113.66 mg CAE g^−1^ DM and was achieved at 80 °C, without compromising antioxidant activity. Results showed that the method employed was a high-performing system providing polyphenol-enriched extract with improved antioxidant characteristics compared with other green solvents.

Rodrìguez-Juan et al. [172] tested the non-polar resin Amberlite XAD-16 to purify PCs from the NaDESs extract obtained from VOO samples. The process was very effective with no losses of yield and solvent recycling. Similarly to the work of Garcìa et al. [61], 14 gr of DES (here, the best choice was ChCl:xylitol:water, with a ratio of 2:1:3) were mixed in a water bath under agitation with 14 gr of VOO samples. The resulting mix was then centrifuged at 1200× *g* for 10 min and the polar layer was recovered. It was washed with methanol, bubbled with nitrogen, and washed with MQ water to finally be injected in the column for the HPLC-DAD analysis. The procedure parameters were also varied to measure the effect of different conditions, as the pH variation. Results showed that acidified water with acetic acid to wash the pretreated column and to set its pH to four allowed a higher total recovery of polyphenols (555.36 mg/kg) compared to washing with non-acidified water (447.08 mg/kg). Target molecules were recovered by sequential elutions performed with 50%, 80%, and 100% ethanol (100 mL each), since the more polar analytes are recovered more with 50–80% ethanol while the less polar ones are eluted only with 80–100% ethanol. Recoveries ranged from 81% to 100%.

Şahin et al. [173] tested 11 DESs containing an HBD (glycerol, ethylene glycol, lactic acid, urea, dimethyl urea, and D-sorbitol) and an HBA (L-proline, citric acid, glycerol, ethylimidazole, and methylimidazole) to extract OLE, verbascoside, and rutin from olive leaves through homogenizer-aided extraction (HAEX) for 60 sec at 13,310 rpm with 48.9% water in the DES. Once the extraction with the DES was completed, the extract solution was filtered (0.45 mm syringe filter) and stored in a refrigerator. Quantification of OLE, verbascoside, and rutin was achieved via HPLC. Among the 11 tailor-designed liquids, citric acid: lactic acid (1:4) showed the superior OLE yield (10.79 mg·g^−1^ DM). Furthermore, the relevant DES had ≈ 8% better yield than that of the ethanol–water solution. However, the best performances were attained by using the methanol–water solution and by using the water extract. Globally, with the HAEX method and the designed DESs, OLE-extracted content varied between 6.45 and 16.31 mg·g^−1^ DM, corresponding to 51.82 and 139.70 mg/L of olive leaf extract. Results also highlighted the fact that OLE extraction was enhanced by the increase in acidity of the medium of the extraction. Indeed, this factor might be preserving the stability of extracts by preventing the oxidation of phenolics due to the enzymatic reactions [174]. Furthermore, the yield started to decrease when the pH of the DES was above six. This outcome is expected since the acidic structure of the PCs might undergo degradation in an alkaline environment [175].

Paradiso et al. [170] made a first attempt to label olive oils according to the minimum amount of HT and its derivatives, such as TY, stated in the EFSA health claim by presenting a simple NaDES-UV method based on liquid–liquid extraction. EVOO samples (0.5 gr) were mixed with 5ml of lactic acid:glucose:water (5:1:3) with 20% water addition (*v*/*v*). After intense vortexing for 5 min, centrifugation was performed for 10 min at 6000 rpm. The lower layer (NaDES plus phenolics) was recovered, centrifuged at 9000 rpm for 5 min and, after being recovered, finally filtered at 0.45 μm using nylon filters. Extracts were analyzed via UV-Vis spectrophotometry. Limits of detection and quantification of total HT plus TY (both free and linked) were, respectively, 3.9 and 11.8 mg kg^−1^. Together with the repeatability, they can be considered satisfactory for screening purposes, thus paving the way for food-grade analytical chemistry.

De Almeida et al. [176] screened different DESs, formed by combining ChCl (as HBA) with either citric, malic, malonic, or acetic acid (HBD species) to extract PCs from olive leaves by conventional heating extraction. The best carboxylic acid-based DES was selected to optimize PC extraction conditions using response surface methodology (RSM). The extraction procedure was carried out in a thermo-shaker (3 h at 50 °C), mixing dried leaves and solvent. Extracts were then diluted and filtered using a PTFE syringe filter with a 0.22 μm pore size (1.3 mm diameter) before UHPLC-MS analysis. Results under optimized conditions showed that ChCl:acetic acid (1:2) with 50% water addition extracted 15% more PCs than ethanol (615.00 mg kg^−1^ DM versus 537.89 mg kg^−1^ DM, respectively) and that also it was more selective, also extracting TY. Ethanol, however, was able to solubilize more OLE than the DES (369.37 mg kg^−1^ DM versus 323.20 mg kg^−1^ DM, respectively). Using the Folin–Ciocalteu method, the DES extracted two times more PCs than ethanol (34.61 mg of GAE·kg^−1^ DM versus 16.03 mg of GAE × kg^−1^ DM, respectively).

Rodríguez-Juan et al. [177] used three different methods to extract oleacein and oleocanthal from 12 blended samples of lampante olive oil (LOO) and 2 blended samples of EVOO. LOO is a low-value olive oil intended for refining and it was once used as a fuel in domestic lighting lamps. It has been proposed that LOO, before refining [178], could be an ideal source for the recovery of oleacein and oleocanthal that are otherwise formed by the malaxation process when OLE and ligstroside are exposed to enzymes—it should be noted that oleacein and oleocanthal are neither present in the leaf nor in the fruit of the olive, so they could be extracted from EVOO but, considering its economic value, that option would give rise to a much more expensive procedure. LOO contains values of 137.5 mg/kg for oleacein and 33 mg/kg for oleocanthal [179] and, considering that it is much cheaper than EVOO, that makes it a more suitable source for the recovery of the two precious secoiridoids. The liquid–liquid extraction procedure that employed DESs made use of ChCl:xylitol:water (2:1:3) and ChCl:1,2-propanediol:water (1:1:1). A total of 14 gr of DES and 14 gr of LOO sample were mixed under agitation in a rotor at 40 °C for 1 h. The mixture was centrifuged at 1200× *g* for 10 min and the polar DES phase was recovered as previously described [171]. Extracts (20 g) were then passed through an Amberlite XAD-16 adsorbent-filled column that was pretreated with ethanol and washed with acidified MQ water. After loading the column, extracts were washed with MQ water, so eluates were collected by ethanol washing. The eluted samples were dried under vacuum at 30 °C, dissolved in methanol, and filtered through a 0.45 μm nylon filter for HPLC-DAD analysis—chromatographic profiles of oleacein and oleocanthal were identified by UV and subsequently confirmed by HPLC-MS analysis—please note that the fractions enriched in oleacein and oleocanthal were re-purified by reverse-phase semipreparative HPLC. Results showed that ChCl:xylitol:water (2:1:3) extraction reached an increment in yield of 30.8% and 61.5% for oleacein and oleocanthal, respectively, compared to those of the conventional method; extraction with ChCl:1,2-propanediol:water (1:1:1) also obtained a higher yield with respect to the conventional method, with increments of 16% and 57.7%, respectively. Similarly, ChCl:xylitol:water (2:1:3) extraction also reached higher yields for oleacein and oleocanthal with respect to the acidified method, with significant increases of 11% and 35.6%, respectively. In regard to the acidified method, instead, ChCl:1,2-propanediol:water (1:1:1) obtained 34% more for oleocanthal but the increase was not significant for oleacein. The results seemed to indicate that the two DES employed, having a polarity closer to methanol than water, have allowed the higher extraction efficiency of less polar compounds as oleacein and oleocanthal present in the olive oil. In this instance, DESs could be a promising green alternative to recover and, subsequently, purify these two secoiridoids from LOO.

Pontes et al. [180] have utilized four ChCl-based DESs with polycarboxylic acids with 16.67% water addition (*w*/*w*) to extract PCs from olive oil pomace. The Brazilian olive pomace was firstly pretreated with a chosen drying method, then dried, milled, protected from light, and stored at −80 °C prior to use. The extraction of PCs was performed in Eppendorf vials in a dry bath under the steer at 50 °C for 3 h. Using the Folin–Ciocalteu method, the extraction potential of the solvents was evaluated based on the quantification of the total phenolic content (TPC). In the study, ChCl:malonic acid (1:1) had the highest extraction potential (19.76 mg of GAE g^−1^ DM) and, in optimized extraction conditions, ChCl:malonic acid (1:1) reached 27.61 mg of GAE g^−1^ DM, which was a value that was 9% above that of methanol (25.32 mg of GAE g^−1^ DM). In addition, it is important to emphasize that when the extraction potential of the ChCl:malonic acid (1:1) is compared to that of water, the DES extracted seven-fold more PCs than water. So, the results clearly showed that ChCl-based DESs with polycarboxylic acids with water addition can be considered to be designer solvents and strong alternatives for PC extraction from olive pomace. Furthermore, the study has appraised the possibility of applying these DESs as an aqueous phase to prepare O/W emulsions, without further purification of extracts.

Yücel and Sahin [181] tested eighteen combinations of carboxylic acids-based DESs to extract OLE as individual compound of some Turkish olive leaves’ extracts through HAE. In particular, the different types (1:1, 1:2, and 2:1) of DESs were designed by combining an HBA (carboxylic acids such as citric and lactic acids) and an HBD (glycerol, ethylene glycol, ammonium, and sodium acetates). The optimized extraction conditions in HAE (90 s of extraction time under 14,000 rpm with 50% water added to the DESs) led, after UV-Vis and HPLC analysis of the extracts, to the highest values of total phenolic content (39.41 mg of GAE g^−1^ DM) and of OLE (14.06 mg g^−1^ DM) that were registered for the best DES, lactic acid:glycerol (1:1).

Akli et al. [182] tested, for the first time, three glycerol-amino acids-based DESs for the extraction of PCs from some Greek olive leaves through a heat and stirring method. DESs were made by mixing glycerol (HBD) with either lysine, proline, or arginine (HBA) in different molar ratios (from 1:1 to 11:1)—the chosen were glycerol:arginine (7:1), glycerol:lysine, and glycerol:proline both at (3:1), while different molar ratios resulted in crystallized mixtures and impracticable DESs. The synthesized DESs were frequently checked over a period of at least one month for their stability and crystal formation. Extraction was carried out in an oil bath under continuous stirring for 150 min. Ethanol, methanol, and water were also used as control solvents. Total polyphenol yield was measured by using the Folin–Ciocalteu method while quantification of the compounds of interest was accomplished through LC-MS/MS. The total phenolic content (TPC) of all DES extracts was found to be higher than that of the control solvents. The extraction with the best performing DES, glycerol:lysine (3:1), allowed for 64%, 66.01%, and 71.09% higher TPC compared to those attained, in the given order, with ethanol, methanol and water. The LC-MS/MS analysis also revealed a selective behavior for the three DESs towards the extraction of TY in respect to the other quantified PCs.

Mir-Cerdà et al. [183] optimized a heat and stirring method to extract targeted analytes form some Spanish olive leaves employing NaDESs based on ChCl (HBA) with three different HBDs (glycerol, urea and lactic acid). Different molar ratios (1:2, 1:5, and 2:1) and water mass percentages of 10%, 20%, and 30% were assayed for the ChCl:glycerol system. Extraction took place by mixing 0.5 gr of sample with 10 mL of NaDESs at 80 °C in a water bath for 2 h under constant stirring. Extracts were also prepared with an EtOH/water mixture as a control. All the extracts obtained were centrifuged at 3000 rpm for 10 min. The supernatant was filtered through a 0.45 μm nylon filter and placed in an HPLC vial. The extracts were stored in the freezer (−18 °C) until analysis and then quantified via LC-UV-MS and LC-UV-MS/MS. Quantitative results showed that Luteolin-7-O-glucoside, OLE, and HT were by far the most remarkable polyphenols in the olive tree leaves under examination extracted with ChCl:glycerol (1:5), with more than 100 mg kg^−1^ of fresh weight (FW) of olive leaves– values ranged between 130 and 260 mg kg^−1^ of the FW.

Carmona et al. [72] set up, for the first time, a three-stage circular method for recycling and re-valorizing fresh OMWW (‘alperujo’) by detoxifying them through NaDESs extraction to produce a dephenolyzed by-product named ‘alpeoNADES‘ that was then bio-transformed (through precomposting and vermicomposting) into a fertilizer. The extraction procedure has foreseen equal amounts (10 gr) of fresh OMWW and NaDESs mixed under the stirrer at 25 °C for 1 h and then centrifugation at 2000× *g*—extraction was repeated twice and the extracts were pooled. For the purpose of the biotransformation, 15 kg of dephenolyzed OMWW were then obtained by mixing equal amounts (300 gr) of fresh OMWW and NaDESs at the same conditions just described. The resulting 15 kg of dephenolyzed OMWW were homogenized and all extracts were filtered through 0.45 μm pores before HPLC analysis. Out of the six formulated and tested NaDESs, which were identified according to the existing literature [184,185], citric acid:fructose (1:1) with 19% water addition led to the highest total polyphenol content (3988.74 mg kg^−1^ of OMWW), followed by glycolic acid:fructose (1:1) with 13% water addition (3509.83 mg kg^−1^ of OMWW) and ChCl:glycerol (1:1) (3190.79 mg kg^−1^ of OMWW): respectively, the NaDES extracted 33.31%, 24.22%, and 16.64% more PCs than control extraction with methanol. Notably, the acid-based NaDESs, i.e., citric acid:fructose and glycolic acid:fructose, gave higher phenolic yields than the non-acidic NaDESs that consisted of glycerol or betaine combined with sugar (sucrose and glucose, respectively) or consisting of two sugars (fructose:sucrose). PCs profiling of fresh OMWW also showed that secoiridoid derivatives (oleacein, glycosides of TY and HT, and, to a lesser extent, simple phenols like HT) were the major components, together with phenyl propanoids, such as verbascoside and oleosides—on the other hand, glycosylated phenols in fresh OMW had been poorly hydrolyzed to aglycons and to simple phenols, allegedly because fresh OMWW did not undergo chemical or microbiological transformation due to storage. The whole methodology applied constitutes a valuable example to enable the transition of the olive oil industry towards a more integrated and circular type of economy design.

### 4.3. Microwave-Assisted Extraction (MAE) of PCs from the Olive Oil Industry

The use of microwaves in the extraction of bioactive compounds has quite recently been postulated as a novel ecofriendly method [113]. Microwave-assisted extraction (MAE), compared to classical Soxhlet extraction or classic heating, for instance, can analyze several samples simultaneously and utilize polar solvents such as MeOH, EtOH, and ethyl acetate that absorb the microwave radiation that is then converted into heat [186,187]—as previously mentioned, the transfer of this energy from the solvent molecules to the cell walls of the plant cells in the reaction medium favors the release of the PCs trapped within the different subcellular compartments; likewise, the same phenomenon is postulated to happen during MAE with DESs/NaDESs as a solvent or a co-solvent. A remarkable example of MAE coupled with DESs for the extraction of PCs can be traced back to 2015 when Wei et al. [188] described the use of different DESs with MAE to obtain four major bioactive compounds of Radix Scutellariae. The following year, Chen et al. presented the MAE-DES methodology for the isolation of bioactive compounds from the Radix Salviae miltiorrhizae [189]. From that time onwards, the study of PCs present in waste materials from the olive oil industry started to appear more consistent. The general attention in this peculiar class of compounds began to arouse interest in their pharmacological properties and, above all, in their noticeable antioxidant scalabilities. Nowadays, scientific effort into making the extraction processes of these compounds progressively more eco-sustainable is spent, fueled by the use of green and eco-sustainable solvents. In this context, DESs/NaDESs for the microwave-assisted extraction of antioxidant compounds have attracted considerable attention, as is demonstrated by the following works.

Another efficient MAE-DESs methodology, by Alañón et al. [190], targeting PCs in olive tree leaves has been examined through the evaluation of the extraction performance of eleven NaDESs based on ChCl in combination with MAE (at 65 °C for 20 min) [162]. DESs were prepared using polyalcohols (1,4-butanediol (1:6), ethylene glycol (1:2), xylitol (2:1), and 1,2-propanediol (1:1)), organic acids (lactic acid (1:2), oxalic acid (1:1), and tartaric acid (2:1)), sugar (maltose) (3:1), and amide (urea)-based DESs (1:2). The study concluded that the best DES was in the class of the alcohol-based: ChCl:ethylene glycol (1:2) with 25% water addition (*w*/*w*) was found to be the optimum HBA-HBD combination to extract most of the phenolics of interest (oleoside, elenolic acid glucoside, hydroxyoleuropein, luteolin glucoside, oleuropein glucoside, OLE, and ligstroside) from olive tree leaves after 16.7 min of irradiation time. It gave similar results to those obtained by the use of conventional methanol/water (80:20, *v*/*v*) with the same technique. Probably for the reason of its linear structure, the mentioned DES can form better interactions with targeted PCs in respect to branched alcohol-based DESs, as opposed to xylitol and 1,2-propanediol that can cause instead more steric hindrance; thus, ChCl:ethylene glycol resulted in better hydrogen bonding and dipole–dipole capabilities. Among the organic acid-based NaDESs, the lactic acid-based NaDES was the most efficient in extracting the targeted PCs. This could be due to the fact that NaDESs composed of oxalic or tartaric acid are highly viscous and hinder the efficiency of extraction due to low mass transport; indeed, these two solvents also showed the worst recovery of extracted phenolics and were discarded. Nevertheless, the NaDES based on tartaric acid was effective in extracting OLE; it was suggested that the extraction of OLE could be affected by the pH of the solvents (the pKa value of oxalic acid and tartaric acid is 1.25 and 2.89, respectively) and recovered efficiently with solvents of not extreme pH values. The sugar based-DES ChCl:maltose was also discarded due to poor extraction efficiency.

An example demonstrating remarkable practicability of DESs on PC extraction with MAE (10 or 30 min at 80 °C) was reported by Bonacci et al. [191]. The authors used five NaDESs; they were ChCl:urea (1:2), ChCl:glycerol (1:1), ChCl:lactic acid (1:1), ChCl:ethylene glycol (1:1) and ChCl: citric acid (1:1). In the first place, the starting material was represented by olive leaves (fresh and dried). The obtained extracts were centrifuged at 1000 rpm for 10 min and the supernatant was collected. The sample was then filtered under vacuum, diluted with ethanol for the successive characterization via HPLC followed by liquid chromatography electrospray ionization quadrupole time-of-flight mass spectrometry (LC-ESI-QTOF/MS). Results were compared to those conducted, according to the same procedure, using water as an extractant. For this control sample, the obtained water extracts were centrifuged and dried under reduced pressure conditions using a rotary evaporator before subsequent analysis. Results of HPLC chromatograms showed an abundant presence of OLE, while the other PCs identified with ultra-performance liquid chromatography (UPLC)-MS were instead present just in traces and so were almost undetectable. Results showed that NaDES with glycerol and lactic acid were the most efficient over the other NaDESs. In particular, the former also had superior OLE extraction capability over water (corresponding dry leaves extract values: 416.08 ± 0.15 ppm versus 174.47 ± 0.42 ppm). The urea-based NaDES and the citric acid-based one were instead discarded due to their low performance. The same procedure was then repeated with ripened olive drupes samples and this time using the five previous NaDESs without and with 20% water addition (*w*/*w*). Using NaDESs without water, the glycerol containing NaDES performed the best, leading to an OLE recovery of 88,320.90 ± 38.03 ppm after only 10 min. In When recurring with NaDESs with water dilution, OLE was unfortunately not recovered. However, all tested NaDESs, apart from the one urea-based, were only able to extract demethyloleuropein and oleacein (3,4-DHPEA-EDA). This time, the best results were given by the diluted ChCl:glycerol NaDES towards demethyloleuropein extraction (869.8605 ± 3.384 30 ppm after 30 min) and by the diluted ChCl:lactic acid NaDES for oleacein recovery (469.912 ± 2.143 ppm after 10 min). Globally, the results taken together show that NaDESs with the lowest viscosity have the best extracting capabilities and, following water addition, their performance even ameliorates.

MAE allows the extraction of bioactive compounds from natural sources using electromagnetic radiation. The advantage of MAE is the rapid increase in temperature resulting in reduced extraction times and a higher extraction yield of the target analytes. A privilege of MAE is the possibility of analyzing several samples simultaneously. On the other hand, one of its disadvantages could be represented by the questionable application in industrial-scale processes due to the high operating costs and the high content of impurities in the obtained extracts. In fact, in response to the intensive extraction conditions, several concurrently extracted analytes require additional purification steps to produce pure extracts [77].

### 4.4. Ultrasound-Assisted Extraction (UAE) of PCs from the Olive Oil Industry

Ultrasound-assisted extraction (UAE) of various different matrices from olive oil has been intensively studied [192]. In several instances, DESs have been combined with ultrasounds in the development of efficient and environmentally friendly extraction protocols.

In one of the very first studies with UAE-DESs, Khezeli et al. [193] used an ultrasonic-assisted liquid–liquid microextraction method based on DESs (UALLME-DES) with ChCl:ethylene glycol (1:2) and ChCl:glycerol (1:2) for extracting target analytes (ferulic, caffeic, and cinnamic acid) from vegetable oils (almond, sesame, and cinnamon oil) and also from olive samples. The RSM and desirability function (DF) allowed the development of an HPLC-UV analytical method that exhibited good linear calibration ranges (between 1.30 and 1000 mg L^−1^), coefficients of determination (r^2^ ˃ 40.9949), and low limits of detection (between 0.39 and 0.63 mg L^−1^). The results were compared with the extraction efficiencies from pure ethylene glycol and glycerol. The relative mean recoveries with DESs ranged from 94.7 to 104.6% and the extraction efficiency decreased more in the presence of ChCl:glycerol (1:2) with respect to ChCl:ethylene glycol (1:2), showing that the latter was a better option. This effect might also depend on the presence of the three hydroxyl groups of glycerol that account for considerable steric hindrances, thus preventing the interactions between the target analytes and the chloride anion. The prepared DESs were used as green extraction solvents for the pre-concentration of target analytes at trace levels.

The extraction with UAE and glycerol-based NaDESs of the total polyphenol and total flavonoid content of some waste agriculture food biomasses was investigated by Mouratoglou et al. [194]. Several abundant agri-food wastes, including lemon peels, OLL, onion solid wastes, red grape pomace, spent filter coffee and wheat bran were used as sources. It was reported that ChCl:glycerol was more efficient compared to sodium acetate:glycerol and sodium potassium tartrate:glycerol NaDESs. However, the quantification of the extraction yield of the total polyphenol content measured in olive leaves revealed that AE was by far the most efficient means. It has also been shown that the extraction efficiency of the NaDESs may be related to their polarity, which can be regulated through combination with water.

Dedousi et al. [195] reported an UAE of PCs from olive leaves in the presence of glycerol:sodium-potassium tartrate:water. The optimization using response surface methodology (RSM) led to 26.75 mg of CAE (caffeic acid equivalents), with a maximum yield of total polyphenols per gr of dry weight. This result was achieved with a 50% (*v*/*v*) aqueous DES, a liquid-to-solid ratio of 45 mL/g and at 73 °C. The DES was equally effective to AM but it displayed inferior antioxidant properties.

A high-efficiency UAE (15, 35, 60 min at 40 °C) prior to HPLC-DAD analysis of the extracts coming from several food wastes was determined by Fernández et al. [102] while, for the first time, considering olive cake, a by-product from the olive oil industry. The evaluated NaDESs were lactic acid:glucose (5:1), citric acid:glucose (1:1), and fructose:citric acid (1:1). They have been chemometrically designed to choose their most appropriate features. Indeed, RSM was used for the optimization of the extraction parameters, including ultrasound time (15, 35, 60 min), sample material/solvent ratio (15, 45, and 75 mg/mL) and water dilution of the optimal NaDESs (0%, 40%, and 75%). Following these chemometrical considerations, citric acid:glucose (1:1) and fructose:citric acid (1:1) NaDESs were discarded, with their two high densities judged as incompatible with HPLC analysis and as a plausible cause of extraction ineffectiveness. Only lactic acid:glucose (5:1) was then selected for further synthetic optimization: 40 °C, 15% of water, and 0.1% (*v*/*v*) formic acid—the stability of PCs is higher in acidic conditions. Regarding analyzed extraction outcomes with PCs from the olive cake matrices, lactic acid:glucose (5:1) has been shown to perform 98% recovery of HT and 109.4% recovery of TY; together with the other results, lactic acid:glucose (5:1) NaDES is capable of dissolving both polar and weak polar compounds compared to conventional solvents, such as methanol and water.

In another study, Chanioti et al. used ChCl as an HBA and they found out that organic acid-based NaDESs (citric acid and lactic acid) were more effective in extracting OLE, HT, and other molecules from virgin olive pomace compared to sugar-based (maltose) NaDESs, polyol-based (glycerol) NaDESs, water, and ethanolic aqueous solvent [101]. To all DESs, water 20% (*v*/*v*) was added. The techniques for the extraction were HAE (at 40 and 60 °C with homogenization speed of 4.000 or 12.000 rpm for 30 min), MAE (at 40 and 60 °C for 30 min), UAE (at 40 and 60 °C for 30 min), and HHPAE (with pressure of 300 and 600 MPa for 5 or 10 min). HAE has been shown to be the most efficient extraction technique at 60 °C with homogenization speed of 12.000 rpm for 30 min. Later, these finding were combined with microencapsulation and nanoemulsion formulations aiming for the protection of the PCs extracted from olive pomace [196]. Also reported was a summary of the studies performed until 2021 with both conventional and innovative techniques in the extraction of PCs from olive pomace samples—HT, vanillin, apigenin, rutin, and luteolin are considered to be the main PCs present in olive pomace. In this regard, the best solvent solutions have been reported as ChCl:caffeic acid (1:2) and ChCL:lactic acid (1:2) coupled with both HAE and UAE (respective Y_TP_s values = 34.08 mg GAE/g; 20.14 mg GAE/g), while ChCL:lactic acid (1:2) afforded 29.57 mg GAE/g and 25.96 mg GAE/g with, respectively, MAE and HHPAE. More recently, the same principles of HAE, MAE, UAE, and HHPAE have been applied to dried ground olive leaves; after HPLC of the extracts, OLE, HT, and rutin have been being identified as the main components to be present, while caffeic acid, vanillin, and luteolin were detectable in only small amounts. In this instance, the combination of choline chloride/lactic acid NaDESs has excelled in HAE, with the highest Y_TP_s value of 55.12 mg GAE/g; however, the Y_TP_s of ethanol extracts were found to be more elevated than those of the NaDESs extracts in most cases. Finally, the maximum antioxidant activity was registered in UAE with ChCl:glycerol [197].

Zurob et al. [109] designed four sugar-based and four organic acid-based NaDESs for the extraction of HT from olive leaves. The NaDESs that exhibited the highest extraction capacity in 74 and 87 ppm range were lactic acid:glucose (5:1) and citric acid:glycine:water (2:1:1). The observed extraction values for HT exceeded four times the concentration achieved in water and two times in the ethanol/water 50:50 (*w*/*w*) mixture. Olives/NaDESs solid-to-solvent 1:4 ratio has rendered the highest extraction efficiency. Additionally, the influence of the pretreatment of olive leaves with cellulase enzymes has been evaluated; however, since it has been proven to only lead to a minimum increment in the extraction of HT, it appears that these solvents are already capable of efficiently dissolving the lignocellulosic fibers of the plant cell wall per sè in large amounts, thus releasing the intracellular contents present in the leaves. COSMO-RS optimization and analysis was undertaken to theoretically support the experimental data and provide deeper understanding of the phenomena associated with the extraction of HT. The theoretical model has proved to be a useful tool for the development of a new NaDESs-tailored system for the extraction of HT.

Plaza et al. [198] developed a combined green method using DESs with SLE, sonication, and supercritical fluids for purifying HT, TY, and OLE from both olive leaves and a semisolid olive waste known in Chile as alperujo [199]. Initially, to set a reference point, Soxhlet extraction was carried out with 5gr aliquots of dried and grounded samples, so the HT, TY, and OLE concentrations in the Soxhlet extracts were determined via HPLC-UV-Vis analysis. Then, a SLE procedure with DESs was separately performed in an ultrasonic bath (40 kHz at 30 °C) by mixing samples and DESs. Extracts were then centrifuged at 1000× *g* × 20 min prior to HPLC-UV-Vis analysis. Results were compared to the same SLE procedure carried out with water and methanol-ethanol. Supercritical CO_2_ was then used as a stripping phase to recover HT, TY, and OLE from the DESs extracts—it also allowed the DESs regeneration by removing them from the high-pressure cell. The optimized procedure (solid waste/DES ratio of ¼ in 120 min ultrasonic bath) guaranteed the best recoveries in DES extracts using ChCl:ethylene glycol (1:2) in respect to ChCl:citric acid (1:1). HT, TY, and OLE extraction efficiencies were calculated considering the ratio between their concentrations in the olive leaf or olive mill waste samples and their concentrations in the DES extracts. The methodology led to good recoveries of HT from the DES extracts. The stability of TY in the liquid phase did not allow its recovery while OLE was recovered by applying supercritical CO_2_ to the mill waste olives extract but not from the leaf olives extract—the lower yield of OLE was attributed to its conversion to HT under the operating conditions. HT better recoveries yields were 81% from olive mill waste DES extract and 57% from olive leaf DES extract, obtained working at 35 °C and 100 bar for the purification step with supercritical CO_2_.

The UAE of HT from olive fruits using seven NaDESs was reported by Liang et al. [200]. In this series, the tailor-made NaDES consisting of a mixture of betaine and malic acid in a molar ratio of 2:1 exhibited the best efficiencies. RSM based on a central composite design was used to optimize the conditions. The best extraction efficiency was observed at the 1:20 sample:solvent mass/volume ratio upon sonication for 33.65 min in an aqueous solution containing 42.13% (*w*/*v*) DES at 43.86 °C. Furthermore, betaine:malic acid was provided to exert essentially no biotoxicity in the model used and it was also found to provide a synergic effect through enhancement of the bacteriostatic activity of HT.

Fanali et al. [201] optimized a method for the further development and validation of the extraction of the most representative PCs from EVOO testing ten DESs based on choline chloride and betaine in combination with different HBDs, with those being six alcohols, two organic acids, and urea. The emulsion was prepared by dissolving 0.5 gr of EVOO with 0.5 mL of hexane and then mixing it with 0.5 mL of DES. After vortexing and centrifugation, the extraction was repeated three times in an ultrasound bath for 20 min. Extracted PCs were separated and detected using HPLC-DAD/ESI-MS. Out of the ten DESs, betaine:glycerol (1:2), with 30% water addition (*v*/*v*) and 1:1 (*w*/*v*) sample-to-solvent ratio, performed as the best solvent. The total concentration of the target compounds was 773.03 and 597.47 μg/g for the selected DES and the mixture of AM, respectively. Recovery values ranged from 75% to 99%.

Ünlü [202] used a UAE technique to extract OLE, caffeic acid, and luteolin from Turkish olive leaves with different NaDESs. Grounded leaves were mixed with NaDES at 55–75 °C for 60 min in a sonication bath. Extracts were then filtered and the clear supernatant used for HPLC-ESI-QTOF-MS analysis. The highest total polyphenol yield was registered for ChCl:fructose:water (5:2:5) as 187.31 ± 10.3 mg of GAE g^−1^ DM and the highest amount of OLE was seen for glucose:fructose:water (1:1:11) as 1630.80 mg·kg^−1^ DM. Taken together, the results showed that NaDESs can be good candidates to be used as an alternative to conventional solvents, as, in fact, the best two performing NaDESs have been found to extract higher quantities of OLE and caffeic acid than methanol.

Morgana et al. [203] used UAE and food-grade NaDESs composed of lactic acid, glucose, and water for the recovery of HT, luteolin, total anthocyanin, and phenols from olive pomace. The NaDESs consisting of lactic acid:glucose:water in a molar ratio of 5:1:9.3 rendered the highest HT content; thus, the bioaccessibility of the extract was investigated during sequential in vitro digestion through exposure to three simulated digesting solutions (oral (OP), gastric (GP), and intestinal (IP)); the total phenolic contents after exposure were put in a graph as µg of GAE per mL of olive pomace extract. The overall high bioaccessibility evaluation of olive pomace–lactic acid:glucose:water NaDES’ extract was demonstrated by the percentage recovery index for the IP solution that was 183% for HT and 75% for luteolin. The stability of the extract after simulated in vitro gastrointestinal process can support the idea of further development of such extracts as novel natural-based bioadditives.

One of the latest studies involving UAE was performed in 2023 by Hu et al. [204]. They reported the application of a series of matrine:panthenol DESs in different molar ratios and water contents for the extraction of HT from OLL. The combination of density functional theory (DFT) calculations and the UAE of olive leaves, with matrine:panthenol (1:4) under optimal conditions, led to the development of an efficient extraction method of HT in up to 4.98 mg/g of OLL. The DES extract exhibited low cytotoxicity and excellent biocompatibility accompanied by anti-inflammatory and bacteriostatic effects at certain concentrations. In addition, the observed antioxidant effects were superior to the water extract.

Seen in parallel with MAE, the extraction of bioactive compounds with UAE has also attracted a great deal of interest and can be regarded, in fact, as an easily performed methodology with improved extraction yields. One of its positive effects can be addressed in the considerable rise in temperature during the extraction process, which is then reflected in the reduction in viscosity and surface tension of the solvents used, thus finally resulting in an overall higher extraction yield. Temperatures that are too high, however, are not recommended due to the plausible degradation or isomerization of heat-sensitive bioactive compounds, which in turn can lead to a hampering of the extraction efficiency. Overall, UAE is considered useful, fast, efficient and less expensive than other innovative techniques; furthermore, the coupling with several probe systems allows the optimization of the main variables for the extraction process.

**Table 2 antioxidants-13-00062-t002:** Extraction methods of PCs using DESs/NaDESs.

DESs-NaDESs				Sample	Instrumental		Operating Conditions	Target Analytes	Key Findings	Total Polyphenol Yield (Y_TP_)	Ref.
Composition	MR	W Content		Type	Extraction Procedure	Analytical Technology					
ChCl:xylitol	(2:1)	25%		VOO	LPME	HPLC-DAD-MS	Water bath at 40 °C with agitation for 1 h (vortexing for 1 min every 15 min); two phases centrifuged at 1200× *g* for 10 min; recovery of upper oil phase.	HT, TY, oleacein, oleocanthal, OLE agylcon, ligstroside aglycon	Increased extraction efficiency for the sugar-based DES over control; data expressed as: mg/kg of PCs extracted; recovery yield % of separated PC in respect to control; area of each peak in the HPLC-DAD chromatogram.	–	[61]
ChCl:1,2-propanediol	(1:1)	–									
ChCl:glycerol	(1:2)	–									
ChCl:sucrose	(1:1)	25%									
ChCl:sucrose	(4:1)	–									
ChCl:1,4-butanediol	(1:5)	–									
fructose:glucose:sucrose	(1:1:1)	–									
ChCl:malonic acid	(1:1)	–									
ChCl:lactic acid	(1:2)	–									
ChCl:urea	(1:2)	–									
ChCl:urea:glycerol	(1:1:1)	–									
glycerol:glycine:water	(7:1:3)	Used as 80% aq. sol.		Dry Greek OLL powder	H&S	LC-DAD-MS	70 °C for 280 min stirring at 600 rpm.	Oleoside, luteolin derivative, luteolin di-glycoside, luteolin rutinoside, quercetin derivative, OLE, OLE isomer, apigenin rutinoside	Increased polyphenol yield compared with conventional bio-solvents, such as aqueous ethanol and water. The DES extract also displayed stronger antioxidant effects.	111.33 mg GAE/g	[161]
glycerol:glycine:water	(7:1:3)	Used as 80% aq. sol.		OLL powder	H&S	LC-DAD-MS	50 °C for 180 min stirring at 600 rpm; centrifugation and collection of the diluted supernatant.	Luteolin glucoside, luteolin glucoside isomer, OLE, apigenin rutinoside	The presence of methyl β-cyclodextrin enhanced the Y_TP_ of the DES/CD (+17.8%) compared with that of AE (Y_TP_ of AE = 95.81 mg GAE/g).	116.58 mg GAE/g	[162]
ChCl:citric acid	(1:2)	30%		Croatian olive pomace and grape pomace	Combined UAE–MAE	HPLC-DAD	Microwave power of 300 W, ultrasound power 50W for 10 min; centrifugation and collection of the adjusted supernatant.	Gallic acid, HT, TY, vanillic acid, vanillin, pinoresinol, catechin from olive pomace and other molecules from grape pomace.	The total polyphenolic content for the olive pomace extracted with the DES was 645.99 (mg kg^−1^ dw), which was higher than those obtained with AE 511.08 (mg kg^−1^ dw).	–	[165]
lactic acid:glucose:water	(3:1:3)	35% father dil.		163 olive oils samples	LLE	UHPLC-UV	Intense vortex agitation; centrifugation; recovery of lower layer; further centrifugation; filtration.	HT and TY derivatives	Alternative method to correctly label the 98.2% of the 163 olive oils samples according to the legal requirements of the EFSA health claim.	–	[166]
octanoic acid:dodecanoic acid	(3:1)	–		OMWW	LLE	HPLC	Concoction agitated in orbital shaker at 25 °C to ensure thorough mixing; water phase sampled and filtered.	TY and endogenous phenols	HNaDESs are a promising food-safe solvent to improve existing processes for the separation of the healthy fraction of polyphenols from the endogenous phenols present in OMWW.	–	[70]
lauric acid:DL-menthol	(1:2)	–									
aprylic acid:DL-menthol	(1:2)	–									
ChCl:glycerol	(1:1.5)	extract:water (1:1.5)		EVOO	H&S-LLE	UPLC-DAD-MS	Magnetic stirring at 25 °C for 15 min; separatory funnel for decantation and phases separation.	HT, TY, oleacein, oleocanthal	The method allowed the quantitative recovery of analytes found in the extracts.	–	[168]
L-lactic acid:glycine	(5:1)	70%	Dried Greek OLL		H&S	LC-DAD-MS; HPLC	Magnetic stirring at 500 rpm, at 50 °C, for 150 min; centrifugation.	HT, rutin, lutein glucosides, apigenin glucoside, OLE, quercetin, apigenin	Higher extraction values with the DES in respect to the controls; the DES yielded to the hydrolysis of OLE to HT.	93.73 mg GAE/g	[169]
L-proline:oxalic acid	(1:4)	50%	Turkish OLL		S	HPLC	Digital homogenization (10,000 rpm for 60 s); centrifugation; filtration.	OLE (also in trace amounts)	The method coupled with further electrochemistry analyses allowed the detection of trace amounts of OLE; the result of the total biophenol concentration was in line with the previous measurements of the group; OLE yield was 15.66 mg GAE/g.	41 mg GAE/g	[42]
L-proline:xylitol	(1:1)	50%								–	
L-proline:sucrose	(1:2)	50%								–	
L-proline:maltose	(1:2)	50%								–	
L-proline:urea	(1:1)	50%								–	
L-proline:glycerol	(1:4)	50%								–	
L-proline:dimethyl urea	(1:1)	50%								–	
L-proline:dimethyl urea	(2:1)	50%								–	
L-proline:malonic acid	(1:1)	50%								–	
L-proline:malic acid	(1:1)	50%								–	
L-proline:N-methyl urea	(1:2)	50%								–	
ZnCl2:glycerol	(1:1)	50%								–	
ZnCl2:dimethyl urea	(1:1)	50%								–	
ZnCl2:dimethyl urea	(2:1)	50%								–	
ZnCl2:malonic acid	(1:1)	50%								–	
ZnCl2:malic acid	(2:1)	50%								–	
ZnCl2:N-methyl urea	(1:1)	50%								–	
dimethyl urea:sucrose	(2:1)	50%								–	
dimethyl urea:maltose	(2:1)	50%								–	
citric acid:glycerol	(1:4)	50%								–	
KCl:urea	(2:1)	50%								–	
imidazole:sucrose	(1:1)	50%								–	
imidazole:malic acid	(1:1)	50%								–	
L-lactic acid:ammonium acetate	(1:1)	70%	Dry Greek OLL powder		H&S	LC-DAD-MS	Stirring at 500 rpm, at 50 °C, for 150 min; centrifugation and collection of supernatant.	OLE, OLE isomer, luteolin glycosides, luteolin rutinoside, apigenin glycoside	The DES at a molar ratio of 7:1 with 54.6% water addition and 0.7% (*w*/*v*) β-CD led to the maximum extraction yield at 80 °C.	–	[170]
L-lactic acid:ammonium acetate	(3:1)	70%								–	
L-lactic acid:ammonium acetate	(5:1)	70%								–	
L-lactic acid:ammonium acetate	(7:1)	54.6%								113.66 mg CAE/g	
L-lactic acid:ammonium acetate	(9:1)	70%								–	
L-lactic acid:ammonium acetate	(11:1)	70%								–	
glycerol:sodium potassium tartrate:w	(7:1:2)	50%								–	
glycerol(aq.) with 7% 2-OH propyl β-CD		60%								–	
AE with 1g/L citric acid at pH 2		60%								60.90 mg CAE/g	
ChCl:xylitol:water	(2:1:3)		VOO		LPME	HPLC- DAD	Water bath at 40 °C under agitation for 1 h; centrifugation; recovery of polar phase.	HT, TY, oleacein, oleocanthal, OLE aglycon, ligstroside aglycone, l-acetoxy-pinoresinol, luteolin, apigenin	The technique led to a higher total recovery of polyphenols (555.36 mg/kg) in the case of prewashing of the column with acidified water. Recoveries ranged from 81% to 100%.	–	[171]
citric acid:lactic acid	(1:4)	From 0 to 100%.	Turkish OLL		HAEX	HPLC	Homogenizer-aided extraction for 60 s at 13,310 rpm; solution filtered.	OLE, verbascoside, rutin	The citric acid:lactic acid DES with 48.9% water addition led to the most efficient extraction of OLE, with an 8% higher performance than 75% ethanol.	–	[172]
citric acid:ethylene glycol	(1:4)										
citric acid:glycerol	(1:4)										
L-proline:lactic acid	(1:8)										
L-proline:ethylene glycol	(1:4)										
L-proline:glycerol	(1:4)										
glycerol:urea	(2:1)										
glycerol:dimethyl urea	(2:1)										
glycerol:D-sorbitol	(8:1)										
ethylimidazole:glycerol	(1:4)										
methylimidazole:glycerol	(1:4)										
lactic acid:glucose:water	(5:1:3)	20%	EVOO		LLE	HPLC- UV-Vis	Intense agitation with a vortex (5 min); centrifugation; recovery of lower layer; centrifugation filtration.	HT, TY, derivatives	Attempt with satisfactory repeatability and limit of detection and quantification values to label olive oils according to the minimum amount of HT and its derivatives, as stated in the EFSA health claim.	–	[175]
ChCl:acetic acid	(1:2)	From 0 to 50%.	Brazilian OLL		H&S	UHPLC-MS	Agitation (3h at 50 °C) at 400–800 rpm according to the optimized conditions; dilution; filtration.	TY, trans-ferulic acid, caffeic acid, OLE, luteolin, kaempferol	The DES with 50% water addition extracted twice the amount of PCs in respect of AE (16.03 mg GAE/g), which, however, still extracted more OLE than the DES.	34.61 mg GAE/g	[176]
ChCl:citric acid	(1:2)									–	
ChCl:malic acid	(1:1)									–	
ChCl:malonic acid	(1:1)									–	
ChCl:xylitol:water	(2:1:3)		LOO, EVOO		LLE	HPLC-DAD	Agitation in a rotor at 40 °C for 1 h; centrifugation; further separtion.	Oleacein, oleocanthal	Both DESs allowed a significant increase in the extraction yield of oleacein in respect to the acidified and conventional methods. However, only the xylitol containing DES gave significant increase in respect to both methods for oleocanthal.	–	[177]
ChCl:1,2-propanediol:water	(1:1:1)										
ChCl:malonic acid	(1:1)	16.67%	Brazilian olive oil pomace		H&S	No further separation of crude extracts.	In Eppendorf vials in a dry bath under the stirrer at 800 rpm at 50 °C for 3 h.	Not specified.	In optimized conditions (66.3 °C), the best DES reached 27.61 mg GAE/g as Y_TP_, which was 9% higher than MeOH (25.32 mg GAE/g).	19.76 mg GAE/g	[180]
ChCl:malic acid	(1:1)	16.67%								10.81 mg GAE/g	
ChCl:acetic acid	(1:2)	16.67%								11.71 mg GAE/g	
ChCl:citric acid	(2:1)	16.67%								9.99 mg GAE/g	
lactic acid:glycerol	(1:1)	50%	Turkish OLL extracts		HAE	HPLC- UV-Vis	90 s of extraction time under 14,000 rpm.	OLE	The best DES—lactic acid:glycerol (1:1)—surpassed the values of Y_TP_s of AM (17.18 mg GAE/g), AE (16.94 mg GAE/g), and water (14.74 mg GAE/g); however, its OLE yield (14.06 mg) was still lower than the one of AM (15.49 mg GAE/g).	39.41 mg GAE/g	[181]
lactic acid:glycerol	(1:2)	50%								14.09 mg GAE/g	
lactic acid:glycerol	(2:1)	50%								15.67 mg GAE/g	
lactic acid:ethylene glycol	(1:1)	50%								15.35 mg GAE/g	
lactic acid:ethylene glycol	(1:2)	50%								15.41 mg GAE/g	
lactic acid:ethylene glycol	(2:1)	50%								14.63 mg GAE/g	
lactic acid:ammonium acetate	(1:1)	50%								9.72 mg GAE/g	
lactic acid:ammonium acetate	(1:2)	50%								11.32 mg GAE/g	
lactic acid:ammonium acetate	(2:1)	50%								11.18 mg GAE/g	
lactic acid:sodium acetate	(1:1)	50%								14.25 mg GAE/g	
lactic acid:sodium acetate	(1:2)	50%								11.25 mg GAE/g	
lactic acid:sodium acetate	(2:1)	50%								14.25 mg GAE/g	
citric acid:glycerol	(1:1)	50%								15.84 mg GAE/g	
citric acid:glycerol	(1:2)	50%								10.77 mg GAE/g	
citric acid:glycerol	(2:1)	50%								11.35 mg GAE/g	
citric acid: ethylene glycol	(1:1)	50%								12.11 mg GAE/g	
citric acid: ethylene glycol	(1:2)	50%								11.91 mg GAE/g	
citric acid: ethylene glycol	(2:1)	50%								11.42 mg GAE/g	
glycerol:arginine	(7:1)	10%	Greek OLL		H&S	LC- MS/MS	Oil bath under continuous stirring for 150 min.	HT, TY, OLE luteolin glucoside, rutin	The glycerol:lysine DES was found as the most effective in extracting the PCs even over conventional solvents; however, selective behavior for the three DESs towards the extraction of TY in respect to the other quantified PCs was measured.	100.01 mg GAE/g 188.39 mg GAE/g 95.96 mg GAE/g	[182]
glycerol:lysine	(3:1)	10%									
glycerol:proline	(3:1)	10%									
ChCl:glycerol	(2:1)	10%	Spanish OLL		H&S	LC-UV-MS and LC-UV-MS/MS	80 °C in a water bath for 2 h under constant stirring; centrifugation; filtration.	24 different PCs but OLE, HT, and luteolin glucoside were the most abundant.	ChCl:glycerol (1:5) with 30% water addition gave the best results. Quantitatively luteolin glucoside, OLE, and HT were present in more than 100 mg/kg fw.	–	[183]
ChCl:glycerol	(1:2)	20%									
ChCl:glycerol	(1:5)	30%									
ChCl:urea	(1:2)	10%									
ChCl:lactic acid	(1:2)	10%									
citric acid:fructose	(1:1)	19%	OMWW		H&S	HPLC-DAD	Stirring at 25 °C for 1 h; centrifugation; homogeniza-tion; filtration.	Oleacein, HT, glycosides of TY and HT, phenyl propanoid derivatives	Three-stage circular method for recycling and revalorizing fresh OMWW. The three best DES significantly exceeded the control extraction with MeOH in the amount of total phenol content expressed as mg/kg of OMWW.	–	[72]
glycolic acid:fructose	(1:1)	13%									
ChCl:glycerol	(1:2)	–									
betaine:sucrose	(2:1)	13%									
glycerol:glucose	(1:1)	21%									
fructose:sucrose	(1:1)	18%									
ChCl:1,4-butanediol	(1:6)	25%	Spanish OLL		MAE	HPLC-DAD-ESI-TOF- MS	65 °C for 20 min; centrifugation and collection of the diluted supernatant after filtration.	Oleoside, elenolic acid glucoside, hydroxy-OLE, luteolin glucoside, OLE Glucoside, OLE, ligstroside	The best DESs gave slightly higher results in respect to AM extraction (Y_TP_ of AM = 23.57 mg GAE/g).	15.68 mg GAE/g 25.00 mg GAE/g 16.22 mg GAE/g 21.06 mg GAE/g 23.92 mg GAE/g 26.61 mg GAE/g 24.10 mg GAE/g 12.10 mg GAE/g 23.75 mg GAE/g	[190]
ChCl:ethylene glycol	(1:2)	25%									
ChCl:xylitol	(2:1)	25%									
ChCl:1,2-propanediol	(1:1)	25%									
ChCl:lactic acid	(1:2)	25%									
ChCl:oxalic acid	(1:1)	25%									
ChCl:tartaric acid	(2:1)	25%									
ChCl:maltose	(3:1)	25%									
ChCl:urea	(1:2)	25%									
ChCl:urea	(1:2)	Both 0 and 20%.	Fresh and dried OLL; ripened olive drupes		MAE	LC-ESI-QTOF/MS	10 or 30 min at 80 °C; centrifugation; collection of supernatant; filtration; dilution.	OLE, dimethyl-OLE, oleacein, traces of other PCs	ChCl:glycerol and ChCl:lactic acid had superior capabilities over the control extraction with water.	–	[191]
ChCl:glycerol	(1:1)										
ChCl:lactic acid	(1:1)										
ChCl:ethylene glycol	(1:1)										
ChCl:citric acid	(1:1)										
ChCl:ethylene glycol	(1:2)	–	Olive oil; other vegetable oils		UALLME	HPLC-UV	Sonication for 5 min in ultrasound bath; centrifugation 10 min at 3000 rpm; recovery of lower phase.	Ferulic acid, caffeic acid, cinnamic acid	Increased extraction efficiency over pure ethylene glycol and glycerol; relative mean recoveries with DESs ranged from 94.7% to 104.6%.	–	[193]
ChCl:glycerol	(1:2)	–									
ChCl:glycerol	(1:3)	10% 10%	OLL; other wastes		UAE	No further separation of targeted analytes.	80 °C for 90 min with sonication power of 140 W, frequency of 37 kHz and AED of 35 W/L; centrifugation and collection of the diluted supernatant.	Only evaluation of extraction efficiencies and of antioxidant activity.	AE was by far more efficient for olive leaves PCs’ extraction.	36.75 mg GAE/g	[194]
sodium acetate:glycerol	(1:3)									34.18 mg GAE/g	
sodium potassium tartrate:glycerol:w	(1:5:4)									27.68 mg GAE/g	
sodium potassium tartrate:glycerol:w	(1:5:3)									–	
glycerol:sodium-potassium tartrate:W	(7:1:3)	Used as 50% aq. sol.	OLL		UAE	LC-DAD-MS-UV-Vis	Sonication bath (140W, 37kHz, AED of 35W/L) for 30 min with temperature monitored; centrifugation; collection of diluted supernatant.	Luteolin and four glycosides thereof, apigenin rutinoside, OLE	The previous test DES with an initial molar ratio of 5:1:4 was adjusted to 7:1:3 and used as an aq. sol 50% (*v*/*v*), with a liquid-to-solid ratio of 45 mL/g and at 73 °C; it was as equally effective as AM but it displayed inferior antioxidant properties.	26.75 mg CAE/g	[195]
lactic acid:glucose—(acidified DES)	(5:1)	15%	Olive cake and other wastes		UAE	HPLC-DAD	40 °C for 60 min with sonication power of 200W frequency of 20 kHz; centrifugation and collection of the diluted supernatant after filtration.	Gallic acid, HT, TY, catechin, caffeic acid, rutin, coumaric acid, trans-ferulic acid, OLE, cinnamic acid, quercetin, luteolin, naringenin, apigenin	The optimized DES with 0.1% (*v*/*v*) formic acid addition allowed 98% recovery of HT and 109.4% recovery of TY and better stability of the PCs in acidic conditions.	–	[102]
citric acid:glucose	(1:1)	–									
fructose:citric acid	(1:1)	–									
ChCl:citric acid	(1:2)	20%	Virgin olive pomace		HAE	HPLC	Different conditions.	HT, TY, OLE, ligstroside, other secoiridoids and other molecules.	ChCl:citric acid was the best solvent for the extraction of PCs with HAE and UAE; ChCl:lactic acid excelled with MAE and HHPAE; HAE allowed for the best extraction efficiency; overall superior efficiency of all techniques over conventional extraction.	values ranging from 13 to 34 mg GA/g dw of pomace	[101]
ChCl:lactic acid	(1:2)	20%			MAE						
ChCl:glycerol	(1:2)	20%			UAE						
ChCl:maltose	(1:2)	20%			HHPAE						
ChCl:citric acid	(1:2)	20%	Dried ground OLL		HAE	HPLC	Different conditions.	OLE, HT, rutin, with traces of caffeic acid, vanillin and luteolin.	ChCl:citric acid was the best solvent for the extraction of PCs with HAE; in most cases, however, the YTPs of ethanol extracts were still found to be more elevated than those of the NaDESs extracts.	55.12 mg GAE/g	[196]
ChCl:lactic acid	(1:2)	20%			MAE					28.80 mg GAE/g	
ChCl:glycerol	(1:2)	20%			UAE					30.17 mg GAE/g	
ChCl:maltose	(1:2)	20%			HHPAE					31.96 mg GAE/g	
lactic acid:glucose	(5:1)		Chilean OLL		SLE-UAE	HPLC-UV	Ultrasonic bath at 35 °C for 24 h; centrifugation; filtration.	HT	The first two DESs had the highest extraction capacity towards HT (74 and 87 ppm range); these values were four times the ones obtained with water and AE.	–	[109]
citric acid:glycine:water	(2:1:1)										
ChCl:fructose:water	(1:1:1)										
ChCl:fructose:water	(1:2:1)										
ChCl:citric acid:water	(1:1:1)	–									
ChCl:lactic acid:water	(1:2:1)	–									
ChCl:glucose:water	(1:1:1)	–									
ChCl:citric acid	(2:1)										
lactic acid:glucose	(5:1)										
citric acid:glycine:water	(2:1:1)										
ChCl:ethylene glycol	(1:2)	–	OLL and semi-solid waste ‘alperu-jo’		SLE combined with supercri-tical CO_2_	HPLC-UV-Vis	Sonication in ultrasound bath at 30 °C and frequency of 40 kHz; centrifugation.	HT, TY, OLE	The methodology led to obtain good PC recoveries from the DES extracts.	–	[198]
ChCl:citric acid	(1:1)	–									
matrine:octanoic acid	(1:1)	30%	Chinese dry powder of olive pulp		Homogenization-UAE	HPLC	Homogeni-zation in a centrifuge tube at 1:20 g/mL, vortexing for 5 min; ultrasound bath (300W and 40kHz) at 30 °C for 30 min; centrifugation; collection of supernatant; filtration.	HT	The best DES (betaine:malic acid) reached an extraction efficiency for HT of 420.7 mg/Kg of olive fruit powder; also, matrine:azelaic acid exhibited excellent extraction performances under the four ultrasonication temperatures examined.	–	[200]
matrine:decanoic acid	(1:1)	30%									
matrine:lauric acid	(1:1)	30%									
matrine:cocinic acid	(1:1)	30%									
matrine:azelaic acid	(2:1)	30%									
betaine:malic acid	(1:1)	30%									
L-carnitine:gallic acid	(2:1)	30%									
betaine:glycerol	(1:2)	From 10% to 70%.	EVOO		LLE-UAE	HPLC-DAD-ESI-MS	Sonication in ultrasound bath at 25 °C with frequenccy of 37 kHz and heating power of 200W for 20 min.	HT, TY, dialdehydic form of OLE aglycon, OLE aglycon isomer, lygstroside aglycon	Betaine:glycerol (1:2) with 30% water addition (*v*/*v*) and 1:1 (*w*/*v*) sample-to-solvent ratio, performed as the best solvent. Recovery values of the targeted PCs ranged from 75% to 99%.	–	[201]
betaine:lactic acid	(1:2)										
betaine:urea	(1:2)										
betaine:ethylene glycol	(1:2)										
betaine:tryethylene glycol	(1:2)										
ChCl:glycerol	(1:2)										
ChCl:lactic acid	(1:2)										
ChCl:urea	(1:2)										
ChCl:ethylene glycol	(1:2)										
ChCl:tryethylene glycol	(1:2)										
ChCl:fructose:water	(5:2:5)		Turkish OLL		UAE	HPLC-ESI-QTOF-MS	Sonication bath (55–75 °C) for 60 min, at a sonication power of 140 W, frequency of 37 kHz, and AED of 35 W/L.	OLE, caffeic acid, luteolin	Glucose:fructose:water (1:1:11) led to a higher recovery of OLE and caffeic acid than MeOH.	187.31 mg GAE/g	[202]
ChCl:glucose:water	(5:2:5)									-	
ChCl:sucrose:water	(4:1:4)									-	
ChCl:lactic acid	(1:2)	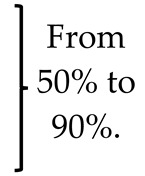								124.05 mg GAE/g	
ChCl:malonic acid	(1:1)									-	
ChCl:ethylene glycol	(1:2)									99.45 mg GAE/g	
ChCl:glycerol	(1:2)									-	
glucose:fructose:sucrose:water	(1:1:1:11)									-	
glucose:fructose:water	(1:1:11)									122.47 mg GAE/g	
glucose:sucrose:water	(1:1:11)									-	
lactic acid:glucose:water	(5:1:9.3)		Lyophilized Argen-tine olive pomace		Homogenization-UAE	HPLC-UV	Homogeni-zation in a centrifuge tube at 75 mg/mL, vortexing during 15 s; ultrasound bath (power of 200 W, 20 kHz frequency) during 60 min at 40 °C; centrifugationand collection of supernatant; filtration.	Luteolin, HT	The first DES rendered the highest HT content; the bioaccessibility of the best extract was investigated during sequential in vitro digestion; values after exposure to the three simulated digesting solution were put into a graph as GAE µg/mL of olive pomace extract.	–	[203]
ChCL:citric acid:water	(1:1:2.7)										
ChCL:levulinic acid	(1:2)										
matrine:panthenol	(1:1)	30%	OLL powder		H&S-UAE	HPLC-UV	Ultrasonic cleaning machine at power of 300, at 30 °C for 60 min; centrifugation; collection/dilu-tion of supernatant; filtration.	HT	The four DESs, under optimal conditions, led to a maximum HT yield of 4.98 mg/g; the possible mechanism of interaction between DES-HT was studied by FTIR, NMR, and DFT calculation; the study presented an alternative method for UAE-based HT extraction.	–	[204]
matrine:panthenol	(1:2)	30%									
matrine:panthenol	(1:3)	30%									
matrine:panthenol	(1:4)	30–60%									

## 5. Results and Discussion

This work can be handy in giving a general idea of the extraction procedures of polyphenols from different matrices of the olive oil industry (OLL, drupes, olive oil, etc.) using DESs/NaDESs coupled with innovative extraction techniques. The following table lists all the extraction procedures reported in this review, while also quoting the main results obtained according to the particular extraction method used under its specific experimental conditions. As a summary and together with other results in the literature, it can be useful in making a quick comparison and aid in orienting towards the procedure that might lead to a good performance.

## 6. Conclusions

In this manuscript, a collection of eco-sustainable extraction methods of antioxidant compounds using DESs/NaDESs were reported. We mostly focused on the PCs obtainable from the industry olearia extracted through non-conventional heating techniques, such as microwave and ultrasound. The articles that have been summarized in this review, with their main key findings, provide valuable insights into the selection of the most effective DESs/NaDESs, in terms of high extraction efficiency, improved selectivity and optimized extraction protocols. The selected DESs/NaDESs appear to be a promising alternative to conventional solvents, mainly based on MeOH and MeOH-H_2_O mixtures, due to enhanced ability to solubilize valuable natural antioxidants. Further investigations, including an in-depth study of the toxicity of the DES extracts, are warranted to fully exploit their potential for cosmetic, pharmaceutical, or food formulations and to establish their commercial viability.

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
