# Peer review of "Alternative Assisted Extraction Methods of Phenolic Compounds Using NaDESs"

_antioxidants, 2023, doi:10.3390/antiox13010062_

Round 1
Reviewer 1 Report
Comments and Suggestions for Authors
This manuscript is about a review of different methods of extraction using naDES, and I have some comments:
- Pag 3, when you said DES are molten salts seems strange. When you have type V DES, non-ionic, they are molten salts? Think about it
- I think it is better you do not classify nades in any DES type, they are disperse
- When you put in the same phrase: hydrophobic deep eutectic solvents (HDESs) and hydrophobic natural deep eutectic solvents (HNaDESs) which are based on quaternary ammonium salts with long alkyl chain; this is wrong. Were Quaternary ammonium salts natural?
- Please include Reference of Figure 1, 2, and Table 1
- Line 178, Van der Waals
- Line 185, missing parenthesis
- All the information in item 4 should be described as table
Author Response
Review 1
This manuscript is about a review of different methods of extraction using NaDES, and I have some comments:
Question: Pag 3, when you said DES are molten salts seems strange. When you have type V DES, non-ionic, they are molten salts? Think about it
Answer: We thank the reviewer for this comment. In the line 103, we reported “DESs (type I-IV) are molten salts”. In the line 105 it is also said “type V of DESs are not necessarily made salts but rather by molecular substances”.
Question: I think it is better you do not classify nades in any DES type, they are disperse
Answer: We thank the reviewer for this comment. The line 75 now says “Furthermore, their natural equivalents, i.e., NaDESs and HNaDESs”. Line 119 more explicitly defines: “Natural” DESs (NaDESs) are instead considered as being DESs derivatives.
Question: When you put in the same phrase: hydrophobic deep eutectic solvents (HDESs) and hydrophobic natural deep eutectic solvents (HNaDESs) which are based on quaternary ammonium salts with long alkyl chain; this is wrong. Were Quaternary ammonium salts natural?
Answer: We kindly thank the reviewer for this comment. Indeed, now line 69 only puts together DESs and HDESs. Then, following line 75, it is spoken about their natural equivalents, i.e., NaDESs and HNaDESs.
Question: Please include Reference of Figure 1, 2, and Table 1
Answer: We thank the reviewer for this comment. We included reference of Figure 1 and Table 1. The Figure 2 has been replaced with a different and higher resolution.
Question: Line 178, Van der Waals
Answer: We thank the reviewer for this comment. We reported the correct name.
Question: Line 185, missing parenthesis
Answer: We thank the reviewer for this comment. We deleted the wrong parenthesis.
Question: All the information in item 4 should be described as table
Answer: We thank the reviewer for this comment. Now Table 2 puts together the main data of the experiments of item 4 which are relevant to our discussion.

Reviewer 2 Report
Comments and Suggestions for Authors
* In the abstract, include more information about the research results of the manuscript
* In the introduction, add more applications with their features of using
NaDESs
* Place more recent works on the problem that is being raised in the introduction
*Figure 1 is very interesting, please place more discussion about the figure in the manuscript
* Improve the quality of figure 2
*Point 2 Overview of DESs shows an interesting analysis, but it is recommended to add recent works with examples
* In general, an interesting analysis is presented in the various points of the manuscript; it is advisable to add some figures to reinforce the information presented in the manuscript
* The following line is shown in the manuscript The extraction efficiency was also good in comparison of conventional solvent (MeOH/H2O (3:2, v/v)) [158]. Please add a more detailed explanation of this point
* In the manuscript, the conclusions must be worked on, since the lines presented are very few, for the type of work being proposed
* It is recommended to place references in the same format
Author Response
Review 2
Question: In the abstract, include more information about the research results of the manuscript
Answer: We thank the reviewer for this comment. In the abstract, we included more information about the rearch results of manuscript.
Question: In the introduction, add more applications with their features of using NaDESs
Answer: We thank the reviewer for this comment. We added more applications, in the introduction fase.
Question: Place more recent works on the problem that is being raised in the introduction
Answer: We thank the reviewer for this comment. We reported more recent works.
Question: Figure 1 is very interesting, please place more discussion about the figure in the manuscript
Answer: We thank the reviewer for this comment. We reported a more discussion about the Figure 1.
Question: Improve the quality of figure 2
Answer: We thank the reviewer for this comment. We improved the quality of Figure 2 and chanced.
Question: Point 2 Overview of DESs shows an interesting analysis, but it is recommended to add recent works with examples
Answer: We thank the reviewer for this comment. We added other works with others figures.
Question: In general, an interesting analysis is presented in the various points of the manuscript; it is advisable to add some figures to reinforce the information presented in the manuscript
Question: The following line is shown in the manuscript The extraction efficiency was also good in comparison of conventional solvent (MeOH/H2O (3:2, v/v)) [158]. Please add a more detailed explanation of this point
Answer: We thank the reviewer for this comment. We added a more detailed explanation of this point
Question: In the manuscript, the conclusions must be worked on, since the lines presented are very few, for the type of work being proposed
Answer: We thank the reviewer for this comment. We have reported further considerations in the "Conclusion" paragraph
Question: It is recommended to place references in the same format
Answer: We thank the reviewer for this comment. We placed references in the same format

Reviewer 3 Report
Comments and Suggestions for Authors
The authors review alternative assisted extraction methods of antioxidant compounds using NaDESs. The authors focus on extraction techniques, such as microwave assisted extraction and ultrasound assisted extraction in relation to the possibility of better exploiting DESs and NaDESs as plausible extracting solvents of the phenolic compounds present in different matrices from olive oil components, such as virgin olive pomace, olive leaves and twigs, virgin and extra olive oil, olive cake and olive mill wastewaters. The review is very interesting. However, some points of the manuscript should be improved. Specific comments are given below.
1. The authors should offer the full name of DESs, NaDESs, HDESs and HNaDESs when they appear for the first time.
2. The authors should offer the function and structure of phenolic compounds.
3. There are many references in Comprehensive list of literature’s works with DESs used in olive oil PCs’ extraction. The authors should compare the differences between references.
4. The authors should further analyze the merit and drawback of microwave assisted extraction and ultrasound assisted extraction of phenolic compounds from oil industry.
5. The authors should offer the outlook of phenolic compounds.
6. The main text of this paper is focus on the phenolic compounds, the authors should add the difference of antioxidant compounds and phenolic compounds.
7. Please carefully check the manuscript for writing and grammar.
Comments on the Quality of English LanguageMinor editing of English language required
Author Response
Review 3
The authors review alternative assisted extraction methods of antioxidant compounds using NaDESs. The authors focus on extraction techniques, such as microwave assisted extraction and ultrasound assisted extraction in relation to the possibility of better exploiting DESs and NaDESs as plausible extracting solvents of the phenolic compounds present in different matrices from olive oil components, such as virgin olive pomace, olive leaves and twigs, virgin and extra olive oil, olive cake and olive mill wastewaters. The review is very interesting. However, some points of the manuscript should be improved. Specific comments are given below.
Question: The authors should offer the full name of DESs, NaDESs, HDESs and HNaDESs when they appear for the first time.
Answer: We thank the reviewer for this comment. We reported the full names of DESs, NaDESs, HDESs and HNaDESs when they appear for the first time (abstract section).
Question: The authors should offer the function and structure of phenolic compounds.
Answer: The authors offered by Figure 1 the structure of phenolic compounds in olive oil. They also briefly reported the major biological activities of secoiridoids and their derivatives.
Question: There are many references in Comprehensive list of literature’s works with DESs used in olive oil PCs’ extraction. The authors should compare the differences between references.
Answer: We thank the reviewer for this comment.
Question: The authors should further analyze the merit and drawback of microwave assisted extraction and ultrasound assisted extraction of phenolic compounds from oil industry.
Answer: We thank the reviewer for this comment. We analyzed the merit and drawback of microwave assisted extraction and ultrasound assisted extraction of phenolic compounds from oil industry
Question: The authors should offer the outlook of phenolic compounds.
Answer: We thank the reviewer for this comment. Figure 3 reports the structures and features of the main phenolic compounds present in olive oil.
Question: The main text of this paper is focus on the phenolic compounds, the authors should add the difference of antioxidant compounds and phenolic compounds.
Answer: We thank the reviewer for this comment. We believe to change the manuscript title. We changed “Alternative Assisted Extraction methods of antioxidant compounds using NaDESs” in “Alternative Assisted Extraction methods of phenolic compounds using NaDESs”.
Question: Please carefully check the manuscript for writing and grammar.
Answer: We thank the reviewer for this comment. We checked the manuscript for writing and grammar.

Round 2
Reviewer 1 Report
Comments and Suggestions for Authors
I just have a last comment: remove the term molten salts in DES definition. It could lead a wrong idea.
Author Response
Dear Review,
We wish to submit an original research article entitled “Alternative Assisted Extraction methods of phenolic compounds 2 using NaDESs” for consideration by Antioxidant
We confirm that this work is original and has not been published elsewhere, nor is it currently under consideration for publication elsewhere.
In this paper, we report several works in which DESs/NaDESs are used as an alternative solvent system coupled to innovative extraction techniques for obtaining small bioactive compounds present as microconstituents in several matrices from olive oil components in good yields. This review is significant because it testifies that there is a growing interest, both in the industrial and scholar sector, in the utilisation of these two types of green solvents in respect of more classical extraction techniques usually performed with conventional solvents.
Please note that in this last reviewed version, we have applied several modifications. In particular, we have thought that it would have been better to put and present under the corresponding paragraphs the experiments that we have chosen from the literature without generating unwanted duplications. These major revisitations are noticeable in section four. Indeed, we have listed in chronological order by year the works we have described. Under paragraph 4.2 there are almost only works done with conventional extraction techniques, while MAE experiments have been shifted to paragraph 4.3. Similarly, paragraph 4.4 contains now experiments performed using UAE methodologies. Conformingly, table n.2 reports the experiments with the same reformulated order of section 4.
Review report 1 (Round 2)
Question: I just have a last comment: remove the term molten salts in DES definition. It could lead a wrong idea.
Answer: we thank the reviewer for the comment. The misleading adjective ‘molten’ has been successfully removed. Furthermore, next to the definition of the acronyms DESs and DELs, we have added the acronym LTTMs, since it appears that some authors prefer to use this other one to underline the types of eutectic mixtures we have described.

Reviewer 3 Report
Comments and Suggestions for Authors
The authors have addressed the problem very well, and the manuscript can be accepted in the present form.
Comments on the Quality of English LanguageMinor editing of English language required
Author Response
Dear Review,
We wish to submit an original research article entitled “Alternative Assisted Extraction methods of phenolic compounds 2 using NaDESs” for consideration by Antioxidant
We confirm that this work is original and has not been published elsewhere, nor is it currently under consideration for publication elsewhere.
In this paper, we report several works in which DESs/NaDESs are used as an alternative solvent system coupled to innovative extraction techniques for obtaining small bioactive compounds present as microconstituents in several matrices from olive oil components in good yields. This review is significant because it testifies that there is a growing interest, both in the industrial and scholar sector, in the utilisation of these two types of green solvents in respect of more classical extraction techniques usually performed with conventional solvents.
Please note that in this last reviewed version, we have applied several modifications. In particular, we have thought that it would have been better to put and present under the corresponding paragraphs the experiments that we have chosen from the literature without generating unwanted duplications. These major revisitations are noticeable in section four. Indeed, we have listed in chronological order by year the works we have described. Under paragraph 4.2 there are almost only works done with conventional extraction techniques, while MAE experiments have been shifted to paragraph 4.3. Similarly, paragraph 4.4 contains now experiments performed using UAE methodologies. Conformingly, table n.2 reports the experiments with the same reformulated order of section 4.
Review report 1 (Round 2)
Question: I just have a last comment: remove the term molten salts in DES definition. It could lead a wrong idea.
Answer: we thank the reviewer for the comment. The misleading adjective ‘molten’ has been successfully removed. Furthermore, next to the definition of the acronyms DESs and DELs, we have added the acronym LTTMs, since it appears that some authors prefer to use this other one to underline the types of eutectic mixtures we have described.
Review report 2 (Round 2)
Question: Minor editing of English language required.
Answer: we thank the reviewer for the comment. Now the whole review has been thoroughly checked for grammatical slips and incoherent sentences.
(IV) If you found it impossible to address certain comments in the review
reports, please include an explanation in your appeal.
We have also slightly adjusted results, discussion and conclusion to provide a better and clearer understanding.
We have no conflicts of interest to disclose.
Please address all correspondence concerning this manuscript to me at monica.nardi@unicz.it
Thank you for your consideration of this manuscript.
Sincerely,
Monica Nardi
